# Single-cell glycolytic activity regulates membrane tension and HIV-1 fusion

Charles A. Coomer[1,2,3,4], Irene Carlon-Andres[1,4], Maro Iliopoulou[1], Michael L. Dustin[5�‡], Ewoud B. Compeer[5�‡], Alex A. Compton[2�‡], Sergi Padilla-Parra[1,4�‡*]

**1** Cellular Imaging Group, Wellcome Centre Human Genetics, University of Oxford, Oxford, United Kingdom, **2** HIV Dynamics and Replication Program, National Cancer Institute, Frederick, Maryland, United States of America, **3** University of Kentucky, College of Medicine, Lexington, Kentucky, United States of America, **4** Division of Structural Biology, Wellcome Centre Human Genetics, University of Oxford, United Kingdom, **5** Kennedy Institute of Rheumatology, University of Oxford, Oxford, United Kingdom

☽ These authors contributed equally to this work.
‡ These authors are joint senior authors on this work.
* spadilla@well.ox.ac.uk

**Data Availability Statement:** All relevant data are within the manuscript and its Supporting Information files.

## Abstract

There has been resurgence in determining the role of host metabolism in viral infection yet deciphering how the metabolic state of single cells affects viral entry and fusion remains unknown. Here, we have developed a novel assay multiplexing genetically-encoded biosensors with single virus tracking (SVT) to evaluate the influence of global metabolic processes on the success rate of virus entry in single cells. We found that cells with a lower ATP:ADP ratio prior to virus addition were less permissive to virus fusion and infection. These results indicated a relationship between host metabolic state and the likelihood for virus-cell fusion to occur. SVT revealed that HIV-1 virions were arrested at hemifusion in glycolytically-inactive cells. Interestingly, cells acutely treated with glycolysis inhibitor 2-deoxyglucose (2-DG) become resistant to virus infection and also display less surface membrane cholesterol. Addition of cholesterol in these in glycolytically-inactive cells rescued the virus entry block at hemifusion and enabled completion of HIV-1 fusion. Further investigation with FRET-based membrane tension and membrane order reporters revealed a link between host cell glycolytic activity and host membrane order and tension. Indeed, cells treated with 2-DG possessed lower plasma membrane lipid order and higher tension values, respectively. Our novel imaging approach that combines lifetime imaging (FLIM) and SVT revealed not only changes in plasma membrane tension at the point of viral fusion, but also that HIV is less likely to enter cells at areas of higher membrane tension. We therefore have identified a connection between host cell glycolytic activity and membrane tension that influences HIV-1 fusion in real-time at the single-virus fusion level in live cells.

## Author summary

Deciphering the mechanism of HIV-1 entry and fusion is crucial to understanding the first stages of infection. Although conceptually the first steps of the HIV-1 fusion reaction

**Funding:** C.A.C. acknowledges support from the Oxford-Cambridge Fellowship Program from the National Institutes of Health. S.P-P acknowledges funding from the Nuffield Department of Medicine Lead-ership Fellowship and all authors from the Wellcome Trust Core Award (203141). Work in the lab of A.A.C. is supported by the Intramural Research Program of the National Insti-tutes of Health, National Cancer Institute, Center for Cancer Research. Wellcome Trust Principal Research Fellowship 100262Z/12/Z supported M.L.D. and European Commission Grant ERC-2014-AdG-670930 supported E.B.C. The funders had no role in study design, data collection and analysis, decision to publish, or preparation of the manuscript.

**Competing interests:** The authors have declared that no competing interests exist.

are relatively well understood, the role of metabolism regulating cholesterol in the plasma membrane and its impact in HIV-1 fusion has not been addressed to date. In this study, we have found a connection between host glycolytic activity and plasma membrane cho-lesterol concentration. These events in turn affect both membrane tension and membrane order which have a direct impact on the HIV-1 fusion reaction.

## Introduction

It is well-established that HIV-1 infects target cells by engaging its Envelope glycoprotein (a trimer of gp160, itself a heterodimer of gp120 and gp41) with host receptor CD4 [1] and co-receptors CCR5 [2] or CXCR4 [3]. Subsequently, the receptor and co-receptor interactions trigger conformational changes in the gp41 transmembrane subunit, facilitating virus-cell fusion and the formation of a post-fusion, thermodynamically-stable six-helix bundle [4]. Identifying elements that underpin the earliest stages of HIV-1 infection is a priority in the search for an HIV cure [5]. Furthermore, numerous studies have highlighted the importance of cellular metabolism in T cell-mediated antiviral responses and control of viral infection [6,7]. However, the role played by single-cell metabolic activity in cellular susceptibility to HIV-1 invasion has not been addressed.

Upon activation, T cells increase rates of glycolysis and oxidative phosphorylation to cope with higher energy demands associated with immune functions [8]. Expression of the glucose transporter Glut1 is increased upon T cell activation and is required for post-entry HIV repli-cation [9]. Moreover, the Glut1-associated increased glycolytic flux has also been observed in HIV-1 infected cells [10,11] and affects virus pathogenicity and production [12]. More recently, it has been shown that basal glycolytic activity of CD4$^+$ T cell subsets correlates with susceptibility to HIV-1 infection [13]. The authors showed that chronic, suboptimal inhibition of glycolytic activity impairs HIV-1 infection in T cells, but it was unclear which stage of HIV-1 infection was affected [13]. Pyruvate made from glycolysis may enter the citric acid cycle to yield biosynthetic intermediates (such as oxaloacetate and citrate) and reduced electron carri-ers (NADH and FADH$_2$). Although in activated T cells most pyruvate is excreted as lactate, increased cytosolic levels of citrate and NAD$^+$ are found [14] suggesting that this pathway is still in use by activated T cells. Citrate is a precursor of cytosolic acetyl-CoA, which is required for cholesterol biosynthesis and a major determinant of cellular membrane architecture [15].

Cholesterol is required for the formation and maintenance of liquid-ordered (Lo) mem-brane domains in addition to reducing membrane tension [16,17]. Higher plasma membrane cholesterol content stretches lipid tails in Lo domains and a discontinuity within the bilayer is created at the phase boundary between Lo and liquid-disordered (Ld) domains, manifesting in line tension. In turn, enhanced membrane tension regulates membrane curvature. Hemifusion (i.e. the merging of contacting outer membrane leaflets without formation of a fusion pore) is regulated by membrane curvature—positive curvature negates the propensity of fusion pores to progress to full fusion, whereas negative curvature facilitates this process [18,19]. Increased cholesterol establishes line tension between Lo and liquid disordered (Ld) domains and gener-ates negative spontaneous curvature in support of fusion, whereas cholesterol depletion increases membrane tension, decreases line tension between Lo and Ld domains and enables positive spontaneous membrane curvature to disfavour fusion [17,20–23].

Membrane order, tension and curvature are significant contributors to several membrane-regulated processes in human cells important for viral infection, particularly enveloped virus entry (i.e. fusion and endocytosis) and egress [24,25]. In addition, boundaries between

cholesterol-enriched Lo and Ld microdomains are well-established determinants of successful entry for a range of viruses, including HIV-1 [26–29]. The phase boundaries between Lo and Ld domains are thought to represent the site of HIV-1 entry, which may be due to line tension derived from Lo and Ld boundaries which drive membrane bending needed to facilitate fusion [21,27]. Intriguingly, it has been demonstrated that plasma membrane cholesterol content of dendritic cells, macrophages, and T cells in HIV-1-infected long-term non-progressors is substantially reduced compared to that of rapid progressors, and that the cellular cholesterol content correlates with the capacity of HIV-1 to spread [30,31]. Therefore, an unexplored relationship may exist between the host cell metabolic state, cholesterol, and virus entry into cells.

A significant development in the study of cellular metabolism is the application of fluorescence-lifetime imaging microscopy (FLIM) in individual cells [32,33]. FLIM provides resolution in single cell microenvironments that are missed by population-based or spectral methods. Here, we observed higher rates of HIV-1 fusion and infection in cells with high glycolytic activity, as reported by their ATP:ADP ratios and intracellular lactate concentrations, compared to cells with low glycolytic activity. Further analysis demonstrated that targeted inhibition of glycolysis by 2-deoxy-d-glucose (2-DG) drastically decreased HIV-1 fusion and infection. Multicolour real-time single virus tracking (SVT) revealed that HIV-1 entry is arrested at the stage of hemifusion in cells that are glycolytically inactive. 2-DG-treated cells displayed a loss of membrane cholesterol and addition of water-soluble cholesterol to these cells rescued HIV-1 fusion. Moreover, 2-DG treatment decreased membrane order and increased tension of cells as measured by FRET-based biosensor probes using FLIM. To further investigate the connection between membrane tension and HIV-1 entry, we developed a novel assay that combines FLIM and single virus tracking techniques to visualize and identify factors pivotal to HIV-1 infection at high-spatiotemporal resolution in single cells. Through this approach, we initially show that HIV-1 particles generate increased flux in global host-cell membrane tension that is, at least in part, CCR5-dependent. Furthermore, we illustrate that HIV-1 requires local decreases in membrane tension during the process of virus-cell fusion, which is prevented during the inhibition of glycolysis, yet rescued via cholesterol supplementation. Overall, our results demonstrate a functional link between the glycolytic activity of the host cell and the success of HIV-1-mediated virus-cell fusion, with an important role for membrane cholesterol establishing a favourable biophysical environment to facilitate virus entry.

## Results

### Basal metabolic state is a determinant of HIV-1 infection in T cells and reporter cells

There is an inherent hierarchy in the susceptibility of CD4 T cell subsets to HIV-1 infection. Recent work observed that overall T cell subset metabolic activity correlates with susceptibility to replication by HIV pseudotyped with VSV-G envelope protein (HIV-1$_{VSV-G}$-eGFP) [13]. Here, we use TZM-bl and MT4 T cells that are homogeneous in metabolic state and have high glycolytic activity similar to activated CD4 T cells susceptible to HIV-1 infection [34]. TZM-bl cells are highly susceptible to HIV-1$_{JR-FL}$ infection, whereas MT4 T cells are highly susceptible to HIV-1$_{NL4.3}$ as well as HIV-1$_{VSV-G}$ infection. Importantly, TZM-bl cells report HIV-1 infection with high sensitivity through a β-Galactosidase-mediated enzymatic activity.

Initially, we observed a dependence of HIV-1 infection on glycolytic activity in MT4 T cells treated acutely (i.e. 2 hours) with 2-DG prior to infection with HIV-1$_{VSV-G}$ (Fig 1A). Additionally, when exposing MT4 T cells to HIV-1$_{NL4.3,}$ we observed a similar inhibitory effect of 2-DG treatment on virus infection (S1A Fig).

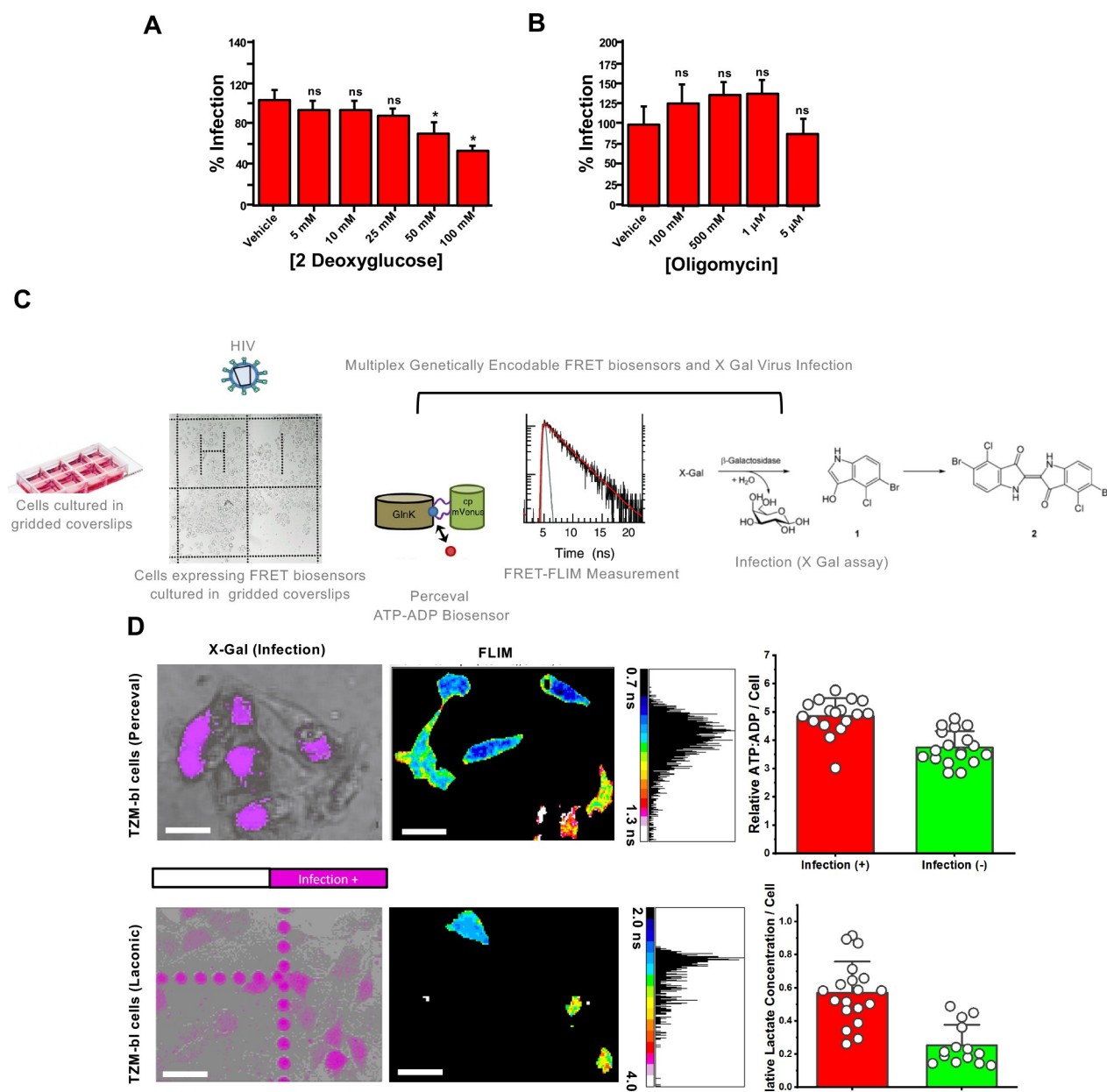

**Fig 1. Relative lactate and ATP/ADP concentrations in single cells correlate with HIV-1 infection.** A.) Bar charts depicting % eGFP-expressing cells (i.e. mean of three independent experiments) as a marker of infection illustrating that acute treatment with increasing concentrations of 2-DG for two hours led to reductions in HIV-1$_{VSV-G}$ infection in human MT4 T cells. B.) Bar charts depicting % eGFP-expressing cells (i.e. mean of three independent experiments) as a marker of infection illustrating that acute treatment with increasing concentrations of oligomycin for two hours did not inhibit HIV-1$_{VSV-G}$ infection in human MT4 T cells. C.) Approach to evaluate if basal, single-cell ATP:ADP ratio could predict HIV-1$_{JR-FL}$ infection in TZM-bl cells. TZM-bl cells were seeded onto an Ibidi 8-well gridded dish and transiently transfected with metabolic sensing biosensors Laconic or Perceval. FLIM images of cells transiently expressing these biosensors were recorded, denoting the location of the cells on the grid. These cells were then treated with HIV-1$_{JR-FL}$ and after 48 hours a β-galactosidase assay was recorded on the same cells as a readout of infection. D.) (Top row) Representative images of β-galactosidase recorded 48 hours after HIV-1$_{JR-FL}$ infection (left) and basal FLIM images of intracellular lactate biosensor Laconic-expressing TZM-bl cells taken before infection (right) illustrating cells with higher intracellular lactate concentrations (warm colours, solid white circle) were more likely to be infected by HIV-1$_{JR-FL}$ (top row, solid white circle); scale bar 50μm. (Bottom Row) Representative images of β-galactosidase recorded 48 hours after HIV-1$_{JR-FL}$ infection (left) and basal FLIM images of intracellular ATP:ADP biosensor Perceval-expressing TZM-bl cells taken before infection (right) illustrating cells with higher intracellular ATP:ADP concentrations (cool colours, solid white 1, 2 circle) were more likely to be infected by HIV-1$_{JR-FL}$ (bottom row, solid white 1,2 and 3 circles); scale bar 50 μm. Built-in negative controls noted with dotted white circles (a, first row and a,b, second row) are presented in both cases (Laconic and Perceval expressing cells). Data depicted in bar charts are a mean of three experiments with at least 30 analysed cells per condition. *** $p < 0.001$ as determined by Student T test of three independent experiments.

Increased citrate and NAD$^+$ levels in activated T cells suggests that glycolysis-derived pyruvate may be used for downstream energy production [6]. Here, we tested whether global reductions in ATP synthesis, rather than the reduction of other biosynthetic intermediates of the TCA cycle, are responsible for the diminished HIV-1 infection observed upon glycolysis inhibition with 2-DG. Treatment with high concentrations of oligomycin, an inhibitor of the F$_0$ proton channel of ATP synthase, failed to reduce HIV-1$_{VSV-G}$ infection in MT4 cells (Fig 1B), illustrating that deficiency in upstream biosynthetic intermediates rather than a global decrease in ATP production leads to reductions in HIV infection.

To determine how a cell's metabolic state correlates with HIV-1 infection, we employed fluorescence lifetime imaging microscopy (FLIM), which enables the quantification of the activity of metabolite reporters at the single cell level. We use FLIM as this methodology provides measurements that are independent of a fluorophore's concentration [35], and increases the reporter's dynamic range (S1B–S1E Fig) [36].

Fig 1C depicts our general strategy to determine if metabolic states could influence virus entry and subsequent infection. Briefly, we transiently transfected Perceval [37], an established ratiometric reporter of the intracellular ATP:ADP ratio, or Laconic [38], an intracellular lactate biosensor, to monitor glycolytic flux in TZM-bl cells via FLIM. As shown in S1B–S1E Fig, treatment of TZM-bl cells with increasing concentrations of 2-DG led to stark changes in the fluorescence lifetime of both Perceval and Laconic in accordance with their biochemical properties, which have previously been described for Perceval (i.e. higher ATP/ADP ratios correspond to lower Perceval lifetimes) [37] and Laconic (i.e. higher intracellular lactate concentrations correspond to higher Laconic lifetimes) [38]. Treatment with 2-DG did not alter the intracellular pH during lifetime measurements, a factor known to distort lifetimes of fluorescent proteins (S2B Fig). Furthermore, two-photon FLIM, a well-established method to decipher the dynamics of NAD(P)H metabolism in living cells [39], showed robust decreases in the ratio of NAD(P)H$_{free}$ vs. NAD(P)H$_{bound}$ in MT4 cells upon 2-DG treatment (S2A Fig, corresponding bar graph in S2D Fig), therefore indicating similar decreases in glycolytic flux upon 2-DG treatment and that this decrease is not exclusive to TZM-bl cells in our study. Treatment of MT4 cells with 5 μM oligomycin did not lead to stark changes in this redox ratio, most likely owing to their pre-activated state (S2C Fig).

Our approach emphasizing fluorescent lifetimes of metabolic biosensors allowed us to calculate relative ATP:ADP ratios or relative lactate concentrations (see Methods) prior to infection, and whether or not these calculated parameters could be predictive of cellular permissiveness to HIV-1 infection, as measured by β-galactosidase activity [40]. Briefly, the β-galactosidase assay utilizes an X-gal stain to detect virus-infected cells as TZM-bls harbour the Tat-responsive reporter LacZ reporter for the detection of Tat-dependent β-galactosidase activities during viral infection. The seeding of cells onto gridded coverslips enabled single cells to be tracked over time. Interestingly, cells that possessed higher relative lactate concentrations (Fig 1D, top row) or a higher relative ATP:ADP ratio (Fig 1D, bottom row) were more likely to become infected by HIV-1$_{JR-FL}$. These results shows that the correlation between glycolytic activity and HIV-1 infection holds true not only during drug treatment of bulk cells but also when accounting for differential baseline metabolic activity of individual cells.

## Basal metabolic state is a determinant of HIV-1 fusion in reporter cells

The separation between HIV-1$_{JR-FL}$-infected and non-infected cells based on their ATP:ADP ratios and relative lactate concentrations in reporter cells prior to infection prompted us to determine the stage at which HIV-1 infection is inhibited. To determine if the preference for HIV-1 to infect glycolytically-active cells is based on pre- or post-fusion determinants in cells,

we multiplexed our FLIM-based reporters into cells and used virus packaging a β-lactamase-Vpr fusion protein (BlaM-Vpr) [41] (Fig 2A), which provides a readout of viral access to the cytoplasm (virus-cell fusion). Briefly, the β-lactamase assay relies on the packaging of the β-lactamase enzyme into nascent HIV-1 virions, where subsequently fusion-positive cells are able to cleave the loaded cytoplasmic FRET-based CCF2 dye, changing the FRET profile readout. Importantly, the BlaM fluorescence profile is not skewed by the ATP:ADP ratio or lactate biosensors because the excitation profiles do not overlap.

Interestingly, single cells with a higher relative lactate concentration were more likely to be fusion positive as determined by the BlaM assay than cells with a lower relative lactate concentration (Fig 2B). Similarly, cells with a reportedly higher ATP:ADP ratio were more likely to be fusion positive than cells with a lower relative ATP:ADP ratio (Fig 2C). Therefore, our data suggests that cells with a higher glycolytic flux are more likely to be infected by HIV-1$_{JR-FL}$, and this bias for glycolytically-active cells occurs at the point of HIV-1 fusion.

## Inhibition of glycolysis blocks HIV-1 fusion at hemifusion

As we initially saw a potential dependence on HIV-1 entry into host cells with host cell glycolytic flux, we tested whether acute arrest of glycolysis via incubation with increasing concentrations of 2-DG would inhibit HIV-1 fusion as reported by the BlaM assay. Treatment of cells with increasing concentrations of 2-DG led to a reduction in both the ATP:ADP ratio and lactate levels in biosensor-expressing cells with minimal loss in cell viability as determined by propidium iodine staining (S1 Fig, S4A Fig).

Pre-treatment of TZM-bl cells with increasing concentrations of 2-DG resulted in dose-dependent decreases in the relative amounts of lactate and ATP:ADP and a concomitant inhibition of HIV-1$_{JR-FL}$ fusion (Fig 3A and 3B). Acute treatment with 100 mM 2-DG blocked HIV-1$_{JR-FL}$ fusion by nearly 80%. Furthermore, we observed similar reductions in HIV-1$_{HXB2}$ fusion in primary CD4+ T cells acutely pre-treated with 2-DG (S3 Fig). Additionally, we similarly observed that increasing concentrations of 2-DG disrupted VSV-G-pseudotyped HIV-1 fusion. Acute glucose deprivation resulted in near-similar reductions in virus-cell fusion events (S4B Fig), although remaining intracellular glycolytic intermediates and glucose at the time of starvation may explain why a more potent inhibition of viral fusion was not reached, as glycolysis may have been allowed to continue briefly after starvation conditions were initiated. Furthermore, acute treatment of TZM-bl cells with 2-DG did not lead to any visible alterations in surface expression of CD4 receptor or CCR5 co-receptor (S4C Fig). Therefore, glycolysis appears to be a major regulator of successful HIV-1 fusion in human target cells.

To further characterise how glycolytic inhibition abrogated HIV-1 fusion in TZM-bl cells, we tracked single virus particles upon exposure to cells. To accomplish this, we co-labelled particles with the lipophilic dye DiD, which incorporates into the viral lipid bilayer, and eGFP fused to Gag to identify the virus core. DiD loss from a particle marked by eGFP-Gag signals lipid mixing between viral and cellular membranes. Furthermore, differences between the surface area of the plasma membrane and endosomal membranes result in differential dilution of the lipophilic dye. As such, lipid mixing at the plasma membrane results in complete dilution of the dye and loss of fluorescent signal, whereas mixing at endosomal membranes has relatively little impact on the lipophilic dye's fluorescent signal (Fig 3C). Irrespective of the initial site of lipid mixing, loss of eGFP-Gag signals that pore formation is complete and cytosolic disintegration of viral core has occurred.

During infection of TZM-bl cells, punctate structures of Gag-eGFP and the lipophilic dye DiD co-localize together at time = 0 sec and subsequently turn red ($t_{1/2}$ = 10.25 minutes). This confirms previous studies showing that HIV-1$_{JR-FL}$ particles enter TZM-bl cells via

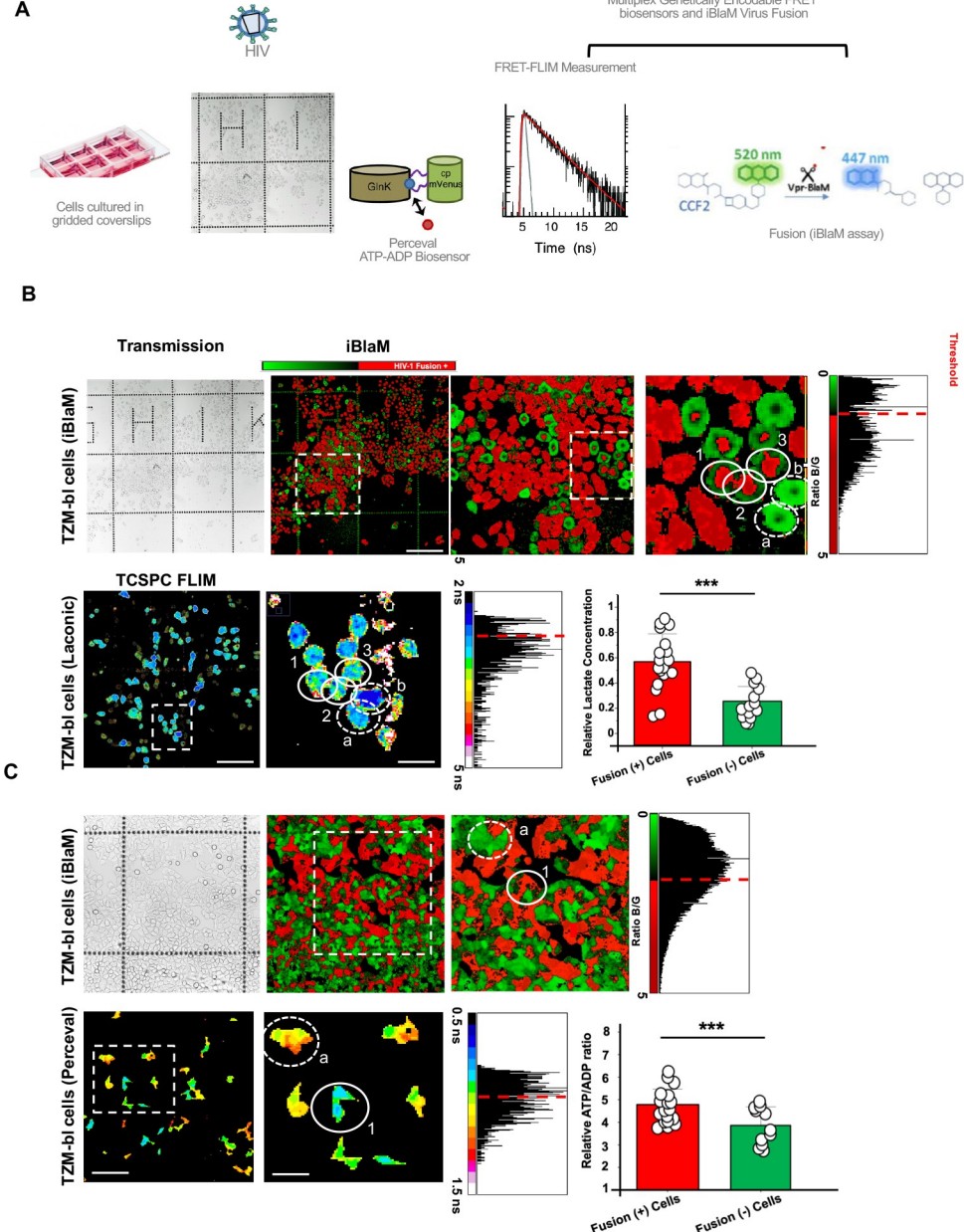

**Fig 2. Relative lactate and ATP/ADP concentrations in single cells correlate with HIV-1 fusion.** A.) Approach to evaluate if basal, single-cell ATP:ADP ratios or intracellular Lactate concentrations could predict HIV-1$_{JR-FL}$ fusion. TZM-bl cells were seeded onto an Ibidi 8-well gridded dish and transiently transfected with metabolic sensing biosensors Laconic or Perceval. FLIM images of cells transiently expressing these biosensors were recorded, denoting the location of the cells on the grid. These cells were then treated with HIV-1$_{JR-FL.}$ After 90 minutes, viruses were washed away and a β-lactamase assay was recorded on the same cells as a readout of HIV-1$_{JR-FL}$ fusion. B.) Representative images of CCF2-loaded cells in Ibidi 8-well gridded dishes recorded 90 minutes after HIV-1$_{JR-FL}$ infection (top row, white solid circles denoting red cells undergoing fusion) and basal FLIM images of intracellular lactate biosensor Laconic-expressing TZM-bl cells taken before HIV-1$_{JR-FL}$ treatment (bottom row) illustrating cells with higher intracellular lactate concentrations (warm colours, solid white circles) were more likely to be scored fusion-positive for HIV-1$_{JR-FL}$ fusion by the β-lactamase assay (bottom row); scale bar, in order of appearance: 200μm, 100μm, 50μm. Built-in controls of cells not undergoing fusion that in turn presented lower lifetimes (cold colours) are stressed with dotted circles C.) Representative images of CCF2-loaded cells in Ibidi 8-well gridded dishes recorded 90 minutes after HIV-1$_{JR-FL}$ treatment (top row) and basal FLIM images of intracellular ATP:ADP biosensor Perceval-expressing TZM-bl cells taken before HIV-1$_{JR-FL}$ treatment (bottom row) illustrating cells with higher intracellular ATP:ADP concentrations (cool colours, solid white circle) were more likely to be scored fusion-positive for HIV-1$_{JR-FL}$ fusion by the β-lactamase assay (bottom row, solid white circle stressing a red fusion positive cells) scale bar, in order

of appearance 100μm, 50μm. Data depicted in bar charts are a mean of three experiments with at least 30 analysed cells per condition. *** p<0.001 as determined by Student T test of three independent experiments.

endocytosis, since Gag-GFP signal is lost while DiD signal (red) persists as it remains in the endosomal membrane (S4D Fig) [41,42]. In contrast, in TZM-bl cells treated with 2-DG (100 mM), double-positive (yellow) particles turn green over time (Fig 3C), indicating DiD dilution at the plasma membrane in the absence of viral content release (Gag-GFP). Specifically, punctate structures of Gag-eGFP and the lipophilic dye DiD co-localize together at time = 0 sec and turn green with a $t_{1/2}$ = 2.53 minutes). This process is suggestive of the hemifusion intermediate preceding full dilation of the fusion pore [42–44]. Moreover, the kinetics of lipid mixing at the plasma membrane corroborates previously reported analyses of hemifusion events [43]. Interestingly, this analysis uncovered two distinct types of DiD disappearance events. One population exhibited a nearly-asymptotic loss of DiD fluorescence (Fig 3C), whereas another population had a slow DiD fluorescence decay lasting up to 10 minutes or longer (S4E Fig). We observed that 7% of single virions rapidly released their lipophilic marker into the plasma membrane, which is similar to previously reported SVT analyses of these events [43]. In summary, these data suggest that cells treated with increasing concentrations of 2-DG display reduced levels of fused virus particles in reporter TZM-bl cell lines and that this arrest to fusion occurred at the hemifusion stage.

In previous studies we [45] and others [46,47] evaluated the impact of spinoculation during HIV-1 entry and fusion and its subsequent influence on directing HIV-1 entry to endosomes. To evaluate this potential influence, we conducted SVT experiments with and without spinoculation (S5 Fig). Our results illustrate that, in non-spinoculated TZM-bl cells, both plasma membrane fusion and endosomal fusion may occur, yet when applying spinoculation protocols, we did not observe plasma membrane fusion events. All of our SVT experiments were performed without spinoculation protocols, enabling the subsequent analysis of membrane properties involved in viral fusion at the plasma membrane and in endosomes (see below). Therefore, the observed hemifusion arrest by 2-DG, in which DiD dye dissipates but Gag-GFP does not, is most likely occurring at the plasma membrane.

## Single-cell metabolic state is linked to plasma membrane cholesterol content

Several studies have linked the inhibition of a variety of metabolic pathways to the availability of lipids in the plasma membrane [48–50], and several of which, including cholesterol, are important for successful virus entry [26,29–31]. To assess whether the inhibition of glycolysis was tied to the amount of plasma membrane cholesterol, we stained for cholesterol with Filipin III after treating cells acutely with increasing concentrations with 2-DG (see Material and Methods). Filipin has been shown to report cholesterol content in cellular membranes. Indeed, in both TZM-bl and MT4 cells, we noticed that acute treatment with 100mM 2-DG led to reductions in cholesterol by approximately 20% and 50% in T cells and reporter cells, respectively, as measured by flow cytometry (Fig 4A and 4B). Immunofluorescence imaging further confirmed these results, illustrating substantial loss of cholesterol from the plasma membrane surface (Fig 4A). Furthermore, treatment with 1mM methyl-β-cyclodextrin (MBCD), an acute cholesterol depletion reagent [51], led to similar reductions in cholesterol staining signal as 2-DG (S6A Fig). Interestingly, treatment with increasing concentrations of ATP synthase inhibitor oligomycin did not lead to a significant decrease in plasma membrane cholesterol (S6B Fig). Furthermore, cholesterol depletion did not exclusively occur at the plasma

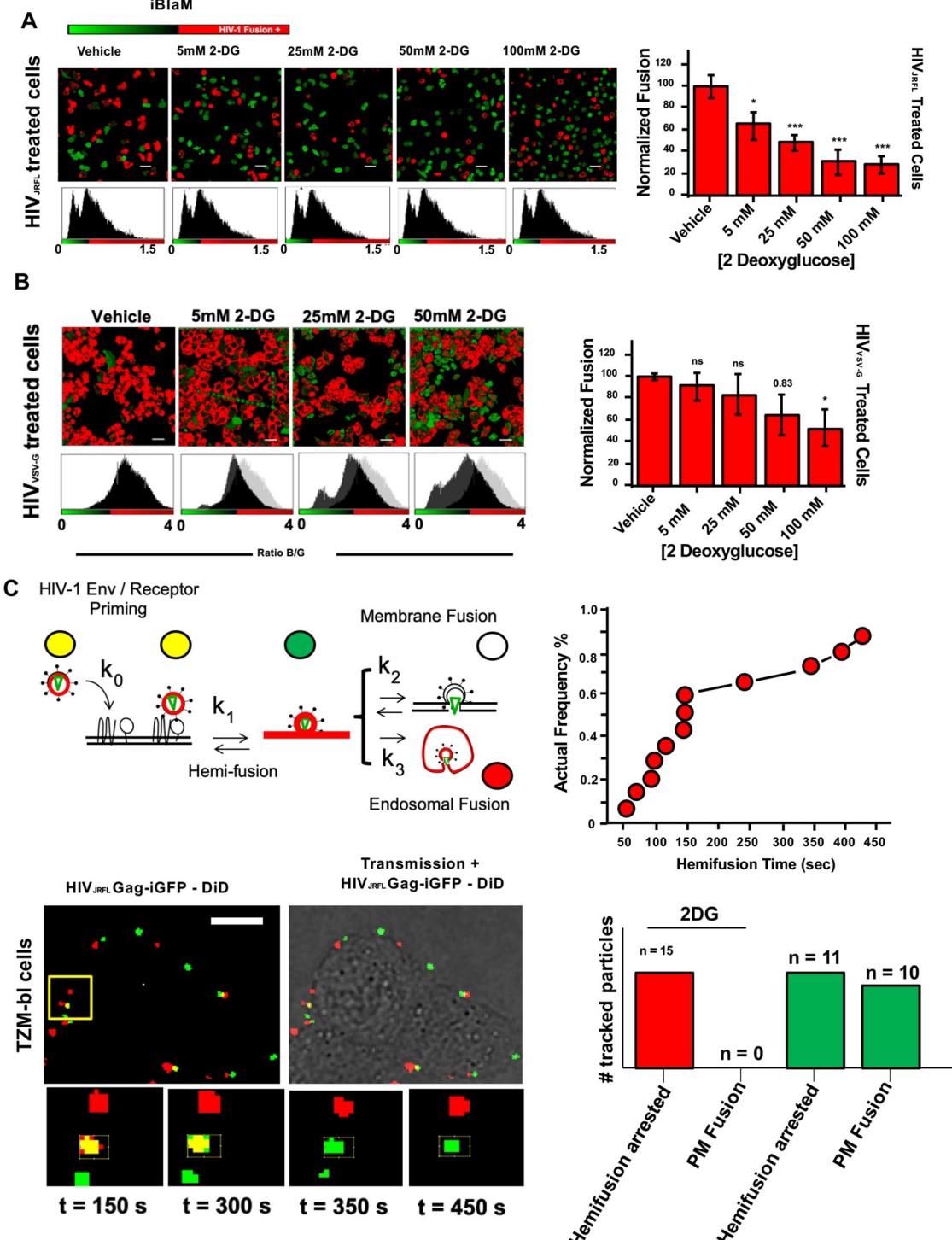

**Fig 3. Addition of 2DG arrests HIV-1 fusion at the hemifusion stage.** A.) (Left) Representative images of CCF2-loaded cells recorded 90 minutes after HIV-1$_{JR-FL}$ infection in vehicle and incrementally increasing 2-DG treatment conditions; scale bar 50µm. (Right) Corresponding bar graph compiling data extracted from the β-lactamase assay and normalised to vehicle-treated control illustrating that increasing concentrations of 2-DG led to reductions in viral fusion for HIV-1$_{JR-FL}$ in TZM-bl cells (mean of three independent experiments). *p<0.5, **p<0.01 *** p<0.001 as determined by one-way ANOVA. B.) (Left) Representative images of CCF2-loaded cells recorded 90 minutes after HIV-1$_{VSV-G}$ infection in vehicle and incrementally increasing 2-DG treatment conditions; scale bar 50µm. (Right) Corresponding bar graph compiling data extracted from the β-lactamase assay and normalised to vehicle-treated control illustrating that increasing concentrations of 2-DG led to partial reduction in viral fusion for HIV-1$_{VSV-G}$ in

TZM-bl cells (mean of three independent experiments). $^{*}$p<0.5, $^{**}$p<0.01 $^{***}$ p<0.001 as determined by one-way ANOVA. C.) (Top row, left) Cartoon diagram illustrating the concept of single-particle tracking with double-labelled virions with DiD and eGFP-gag. Briefly, double-labelled virions entering via endocytosis will have their eGFP-gag signal infinitely diluted during endosomal fusion whilst DiD signal is retained in the endosome, which is mobile. Virions entering via plasma membrane fusion will have their DiD signal infinitely diluted in the plasma membrane whereas the eGFP-gag signal is retained and mobile. Hemifusion is denoted when DiD signal infinitely diluted in the plasma membrane whereas the eGFP-gag signal is retained and immobile. (Top row, right) Kinetics of the individual hemifusion events plotted as cumulative distributions as a function of time. (Bottom row, left) Representative panel of images illustrating doubled-labelled HIV-1$_{JRFL}$ particles losing DiD signal (red) and maintaining immobile eGFP signal (green) when attempting fusion in 2-DG treated cells, suggesting arrest at hemifusion (n = 15, acquired during three independent experiments). (Bottom row, right) Bar chart representing the total number of HIV$_{JRFL}$ double-labelled particles tracked for TZM-bl cells treated with 2DG (red bars) and without treatment (green bars). Only in cells without 2-DG treatment plasma membrane fusion was observed. Total number events tracked in control conditions: 217. Total number of events tracked in 2-DG-treated conditions: 236.

membrane during acute treatment with 2-DG, as endosomal cholesterol content decreased by approximately 40% during acute 2-DG treatment (S6C and S6D Fig). Moreover, 2-DG treatment similarly decreased cell surface cholesterol in primary CD4+ T cells, indicating that our results were not isolated to cell lines (S6E Fig).

To determine if the loss of plasma membrane cholesterol via the inhibition of glycolysis was responsible for the block in virus-cell fusion, we pre-treated our reporter cells with 100mM 2-DG and with increasing concentrations of water-soluble cholesterol to determine if restoration of plasma membrane cholesterol in the face of glycolytic inhibition would restore HIV-1$_{JR-FL}$ fusion in reporter cells. We observed a near-complete rescue of virus fusion when cells were supplemented with 100μg/mL water-soluble cholesterol as reported by the β-lactamase assay (Fig 4C). Our results were confirmed by confocal microscopy of β-lactamase-treated cells. Furthermore, acute treatment of primary CD4 T cells cholesterol-lowering drug Simvastatin, a cholesterol synthesis inhibitor, led to the reduction in HIV-1$_{JR-FL}$ fusion (S3B Fig). Ultimately, it appears that the loss of plasma membrane cholesterol and not reduction in ATP upon inhibition of glycolysis negatively impacts HIV-1$_{JR-FL}$ fusion.

## Single-cell glycolytic activity influences HIV-1 fusion by affecting membrane order and tension

To assess whether inhibition of glycolytic activity alters the membrane order (i.e. a measurement of membrane packing) in single cells, we utilised a recently established sensor for membrane order, FlipTR [52]. FlipTR, a planarizable, push-pull probe with high photostability [52,53] constructed from two large dithienothiophene flippers, segregates with similar efficiency into different membrane phases [54] and maintains the cell's innate membrane order (Fig 5A). We loaded our cells with FliptR and determined the effects of acute glycolysis inhibition on membrane order in TZM-bl cells and compared these effects to cells supplemented with or depleted of cholesterol (Fig 5B). The distribution of lifetimes was homogenous throughout the entirety of the cell regardless of cholesterol enrichment or depletion (Fig 5A and 5B), indicating that the liquid-ordered and disordered phases are mixed in our spatiotemporal resolution, which has been reported previously in similar time-correlated single photon counting (TCSPC) platforms [52,55]. Treatment with 1mM MBCD showed reductions in lifetime by more than 200ps (Fig 5B), indicating a drop in membrane order. Interestingly, cells treated with 2-DG showed similar reductions in membrane order when compared with control cells, supporting that acute 2-DG treatment leads to a loss in high-order lipids in the plasma membrane (Fig 5B). These results were confirmed by comparisons of extracted TCSPC histograms. Interestingly, supplementation of 2-DG treated cells with water-soluble cholesterol capable of plasma membrane insertion restored membrane order lifetime values to vehicle-

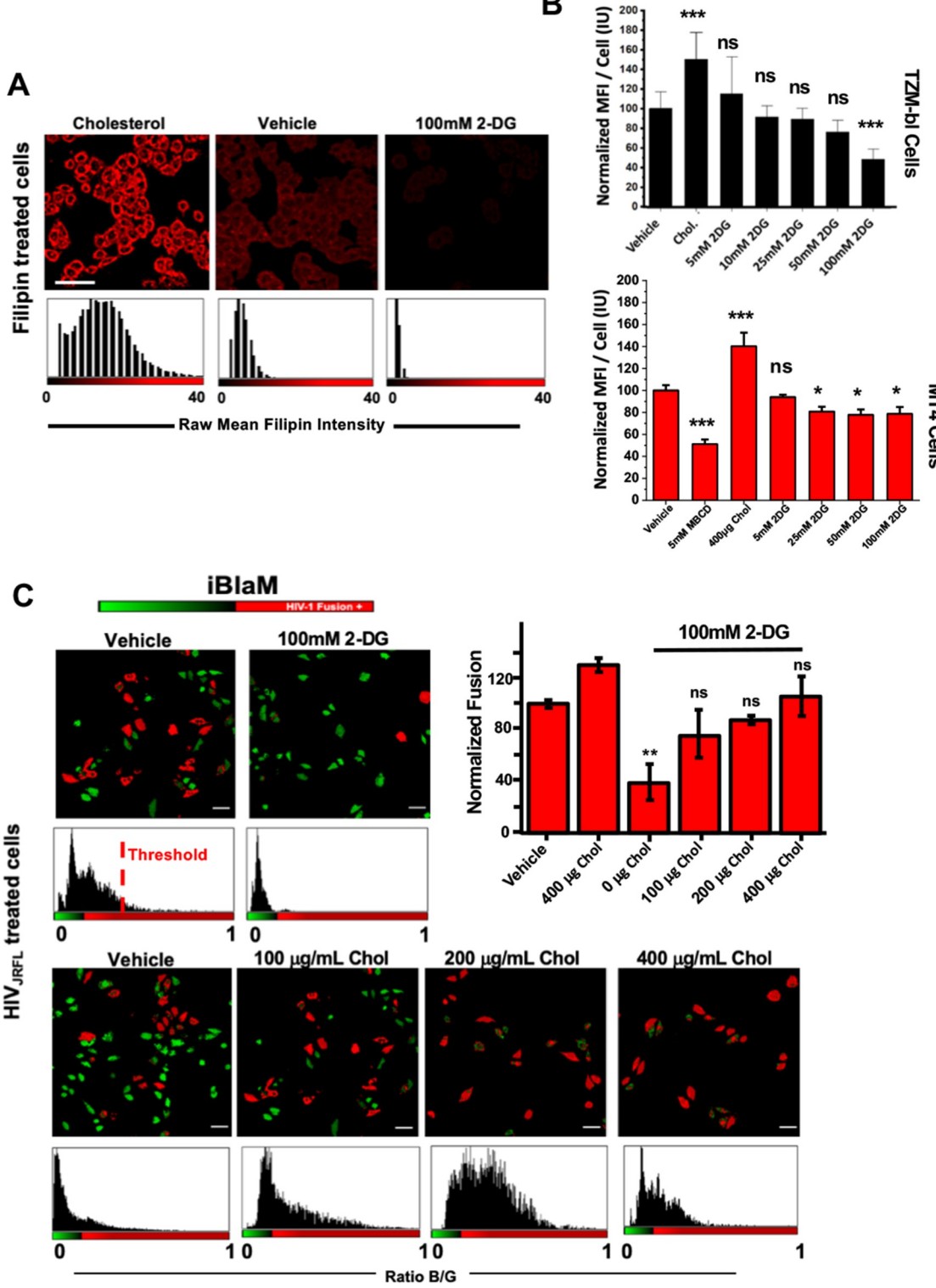

**Fig 4. Addition of 2-DG sequesters cholesterol from the cell membrane.** A.) Representative images depicting filipin mean fluorescence intensity per cell indicating two-hour incubation of increasing concentrations of 2-DG decreases surface cholesterol in TZM-bl cells; scale bar 50 µm. B.) Bar charts depicting mean fluorescence intensity per cell of three independent experiments normalised to vehicle in TZM-bl cells (top) and MT4 cells (bottom) in cells treated with 400µg/mL cholesterol, 5mM MBCD and increasing concentrations of 2-DG. C.) Representative images (left) and bar chart (right) of three independent experiments depicting percent of fusion positive cells, as determined by the BlaM assay, relative to vehicle control illustrating increasing

concentrations of cholesterol rescues fusion in 2-DG treated TZM-bl cells; scale bar 50 μm. *p<0.5, **p<0.01 *** p<0.001 as determined by one-way ANOVA.

treated values. Notably, increasing membrane tension by subjection to hypo-osmotic shock did not lead to a significant change in the membrane order of the cell.

We utilised a previously published FRET-based reporter of membrane tension, MSS [56], to determine changes in membrane tension of single cells during acute glycolytic inhibition. MSS is an ECFP-YPet FRET-based, membrane-bound tension reporter constructed from an

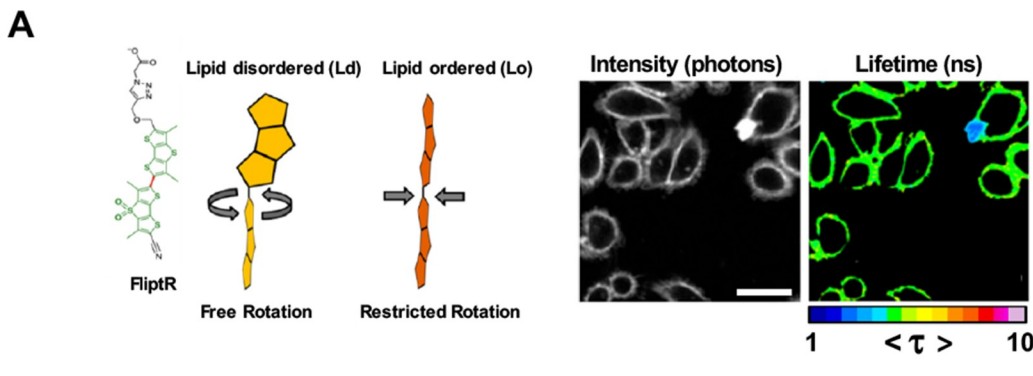

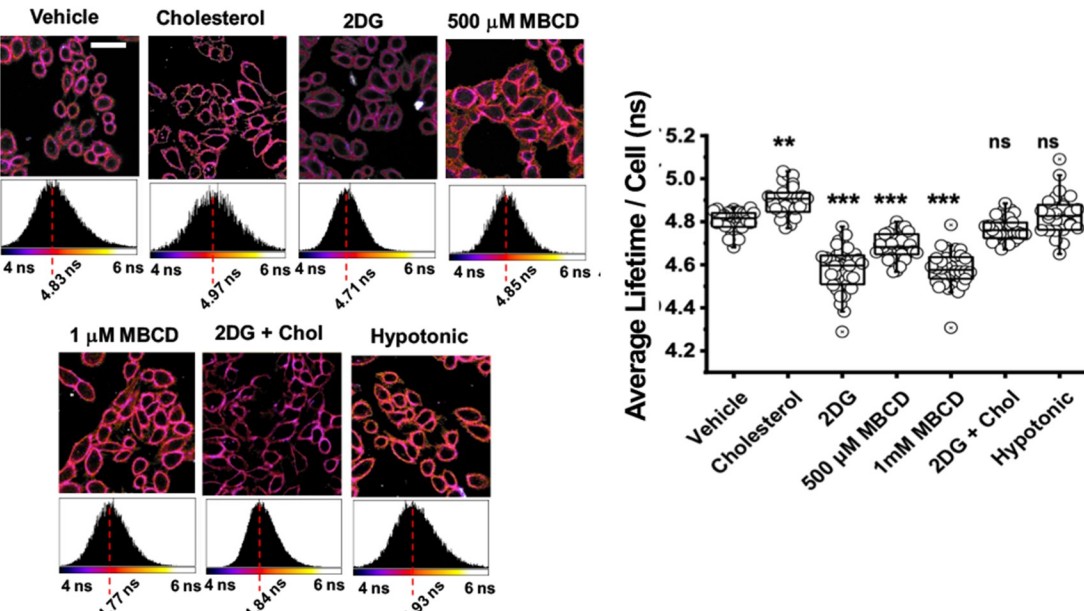

**Fig 5. Addition of 2-DG decreases cellular plasma membrane order.** A.) (Left) Cartoon diagram depicting a model of the planarizable, push-pull membrane order probe FlipTR, constructed from two large dithienothiophene flippers, in Ld and Lo environments. (Right) Representative image of FlipTR-stained cells in untreated conditions depicting the homogenous distribution of lifetimes throughout the entirety of the cell; scale bar 25μm. B.) (Top) Representative images of FlipTR-stained cells in listed treatment conditions depicting the extracted $\tau_m$; scale bar 25μm. (Right) Bar charts depicting the average long lifetime ($\tau_1$) in TZM-bl cells under listed treatment conditions (n = at least 30 per condition, collected during the course of three independent experiments). * p<0.05 ** p<0.01 *** p<0.001 as determined by one-way ANOVA.

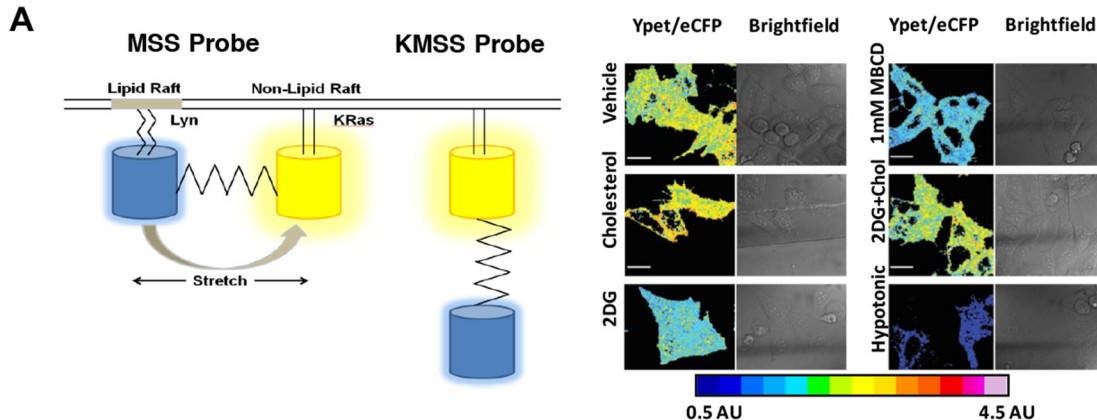

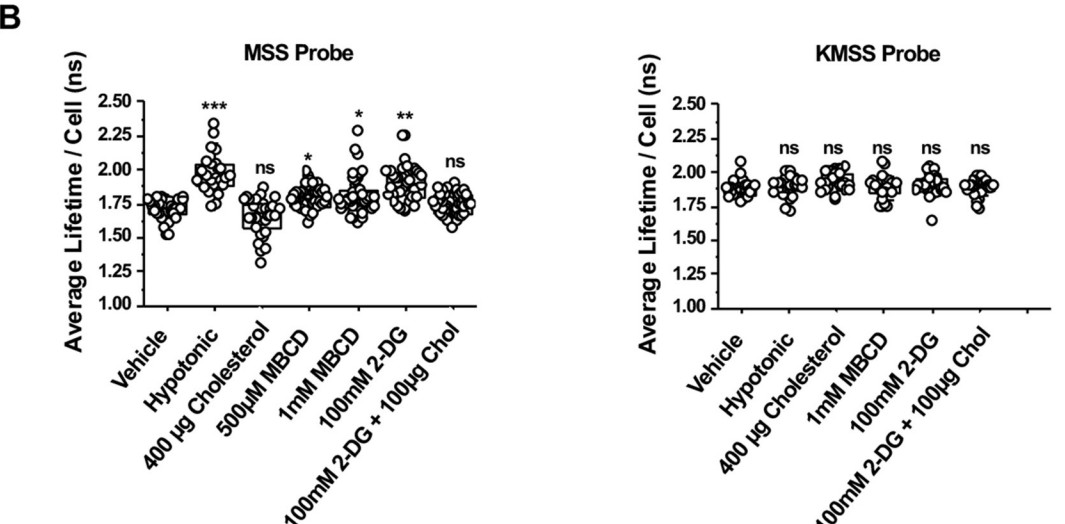

**Fig 6. Acute treatment of TZM-bl cells with 2-DG increases plasma membrane tension.** A.) (Left) Cartoon diagram model depicting membrane tension FRET probe MSS and its control probe (insensitive to membrane tension), KMSS. (Right) Representative pseudocoloured FRET efficiency (i.e. Ypet/eCFP) images of TZM-bl cells expressing MSS under listed treatment conditions; scale bar 10μm. B.) (Left) Dot plots depicting donor CFP lifetimes of the tension-sensitive MSS probes extracted from single cells in the treatment conditions listed (n = at least 20/condition, collected during the course of three independent experiments; *p<0.5, **p<0.01 *** p<0.001 as determined by either one-way ANOVA). (Right) Dot plots depicting donor CFP lifetimes of the tension-insensitive KMSS probes extracted from single cells in the treatment conditions listed (n = at least 30/ condition, collected during the course of three independent experiments; *p<0.5, **p<0.01 *** p<0.001 as determined by either one-way ANOVA).

elastic tension-sensing module and two proteins linked with lipid molecules which anchor in raft and non-raft regions of the plasma membrane (Fig 6A) [56]. Following the same treatment conditions as those used for the FliptR reporter, we detected a similar pattern in changes in CFP lifetime changes and FRET efficiency (i.e. YPet/ECFP ratio) when compared to vehicle-treated cells (Fig 6). Acute cholesterol depletion with mild concentrations of MBCD increased the donor lifetime by more than 100ps (i.e. $\Delta\tau = 101ps$) (Fig 6), indicative of an increase of membrane tension. This result suggests that cholesterol regulates membrane tension in cells. Remarkably, 2-DG treatment led to even larger increases in donor lifetime (i.e. $\Delta\tau = 165ps$), indicating a reduction in FRET efficiency and an increase distance (i.e. stretch and tension) between donor and acceptor fluorophores (Fig 6B). Furthermore, supplementing 2-DG treated

cells with water-soluble cholesterol diminished membrane tension values to control levels. None of the treatments altered donor lifetime values of the KMSS probe (Fig 6B, right panel), which lacks the Lyn domain necessary for membrane association.

Cellular plasma membrane tension is a major regulator of several cellular processes, including those that favour or disfavour virus entry [21,25,27,57]. Since we observed a connection between the glycolytic state of the cell and its corresponding membrane order and tension, we sought to measure how HIV-1$_{JR-FL}$ is impacted by membrane tension during the fusion process. To accomplish this, we analysed single cells transiently expressing the MSS tension reporter during mCherry-Gag-labelled HIV-1$_{JR-FL}$ fusion via single particle tracking concomitant with FLIM of the MSS reporter (Fig 7A). Virus fusion was determined by the rapid loss of mCherry signal. Here, in order to assess HIV-1 fusion, we followed mCherry-Gag labelled HIV$_{JR-FL}$ viruses that spectrally complements the MSS donor / acceptor FRET pair (ECFP/ YPET). The disappearance of the mCherry-Gag signal for viruses sitting in the plasma membrane (labelled with the MSS biosensor) was employed as a proxy for plasma membrane fusion (Fig 7B). We gave up the fourth channel (DiD) as it overlaps with the mCherry signal (both emitting photons in the red region of the spectrum) and employed the signal from the host plasma membrane instead to evaluate membrane tension. Although the average lifetimes for cells expressing MSS probe exposed to HIV particles may recapitulate a common behaviour under different conditions (Fig 7A, right panel), we also analysed the standard deviation (SD) of the average lifetimes fluctuation for all pixels (in time and space) for cells expressing the MSS biosensor. Indeed, in a previous report [58] we have shown that rapid FRET-FLIM (acquisitions in the order of 1–5 sec) gives access to evaluate environmental fluctuations and how they affect protein-protein interactions. We endeavoured to evaluate the amplitudes of the fast average lifetime fluctuations in the same way as in our previous work [58] to gather information about the local environment of single HIV particles when engaged with the plasma membrane of the host.

Interestingly, we initially observed that cells treated with HIV-1$_{JR-FL}$ possessed larger average lifetime fluctuations in CFP donor lifetimes regardless of 2-DG treatment, indicating that cells exposed to virus particles actively undergo changes to membrane tension in response to HIV-1$_{JR-FL}$ (Fig 7A, top right panel). These broad fluctuations in membrane tension were Env-dependent, as non-enveloped HIV-1 particles failed to trigger them during SVT acquisition. Pre-treatment of cells with CCR5 HIV-1 entry inhibitors Tak779, a CCR5 antagonist which inhibits co-receptor engagement by HIV-1 gp120, and T20, an HIV-1 fusion inhibitor which competitively binds to gp41 and blocks its post-fusion structure suggested that these changes were CCR5 dependent. Indeed, MSS CFP-donor lifetime fluctuations in Tak779-treated cells were narrowly similar to non-enveloped, naked HIV-1 virions (Fig 7A). In contrast, upon T20 treatment, these CFP donor lifetime fluctuations were partially restored, confirming that fluctuations in whole-cell target membrane tension during HIV-1$_{JR-FL}$ entry might be CCR5-dependent. These results were confirmed by analysing the pixel-by-pixel lifetime standard deviations of MSS-containing regions of whole cells during virus entry with and without entry or metabolic inhibitor addition (Fig 7A, bottom two panels).

We also evaluated local changes in membrane tension during HIV-1$_{JR-FL}$ entry into individual cells (Fig 7B). We tracked the MSS CFP donor lifetime in 3 x 3 pixel regions at the plasma membrane overlapping with mCherry-Gag labelled HIV-1$_{JR-FL}$ during virus approach by multiplexing our FLIM with SVT. Virus fusion was determined by the rapid loss of mCherry-Gag signal. In reporter cells free from any treatment, we noticed a sharp drop in MSS donor lifetimes in plasma membrane regions coinciding with the disappearance of proximal mCherry-Gag signal (Fig 6B, bottom left panel, green curve). This sharp decrease in donor lifetime values during viral fusion was not noticeable when analysing the lifetimes of the

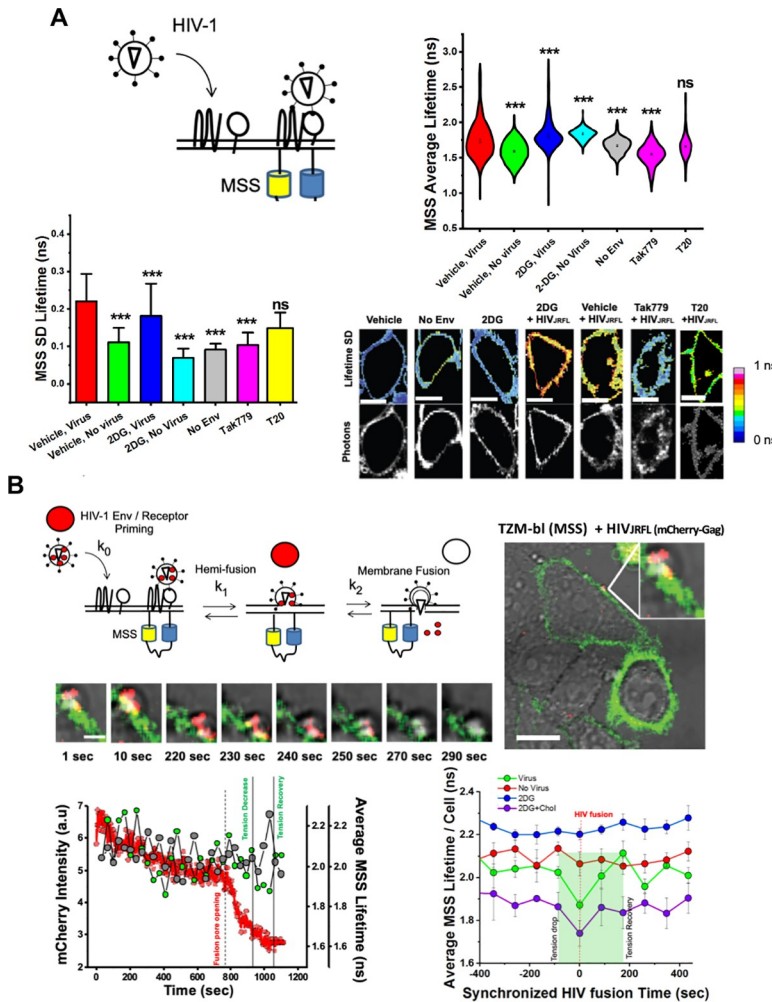

**Fig 7. Multiplexed FLIM with SVT reveals a drop in local tension during single HIV-1 fusion in live cells.** A.) (Top Left) Cartoon diagram depicting approach to determine plasma membrane tension fluctuations during virus entry. (Top Right) Violin plots of average MSS lifetimes per cell per frame during FLIM acquisition during viral entry in TZM-bl cells. Dots in each plot represent the mean of each condition, which was the calculated from at least 50 frames for at least 20 cells per condition during three independent experiments. (Bottom Left) Standard deviation of MSS lifetimes per cell from real-time FLIM acquisition during viral entry in TZM-bl cells, which was calculated from at least 50 frames for at least 20 cells per condition during three independent experiments. * $p<0.05$ ** $p<0.01$ *** $p<0.001$ as determined by one-way ANOVA of three independent experiments. (Bottom Right) Representative images of the standard deviation of MSS lifetimes per cell for each condition during viral entry; scale bar 2μm.* $p<0.05$ ** $p<0.01$ *** $p<0.001$ as determined by one-way ANOVA of three independent experiments. B.) (Top Left) Approach to combine SPT and MSS lifetime values to analyse local membrane tension during HIV-1$_{JR-FL}$ entry. (Right and below) Representative images of a single -mCherry-Gag JR-FL pseudotyped VLP entering MSS-transfected TZM-bl cells are also depicted; scale bar is 20μm and 2μm for the large and smaller images, respectively. (Bottom Left) Representative line-graphs showing local lifetimes in overlapping mCherry-Gag regions (green) and non-overlapping regions (grey) for a single virus particle during entry (loss of mCherry signal, red) in TZM-bl cells. (Bottom Right) Compiled local MSS lifetimes during multiple viral fusion events (green) and respective virus-free regions within the same cell (red) compared to regions of 2-DG treated cells where viruses were incapable of fusion (blue) or rescued events due to cholesterol treatment (purple) in TZM-bl cells. Each point represents a mean of at least 20 cells obtained during three separate experiments.

whole cell (S7A Fig). In comparison, plasma membrane regions lacking virus particles in the same cell did not display any sharp decreases in MSS donor lifetime during virus entry (Fig 6B, bottom left panel, grey curve). The local drop in MSS lifetime (i.e. membrane tension)

overlapping with mCherry-Gag signal disappearance in space and time was statistically significant when compared to mCherry-Gag signal-free regions of similar size in the same cell (Fig 7B, bottom right panel, S7B Fig)). However, there was no statistically significant difference in plasma membrane tension values between these two regions at baseline, before loss of mCherry-Gag signal (Fig 7B, bottom right panel, Su)). Furthermore, this local drop in MSS CFP donor lifetime was transient, as there was rapid restoration of the MSS CFP donor lifetime almost immediately after the loss of the mCherry-Gag signal (Fig 7B, bottom right panel, green curve). In fact, of the total events analysed (n = 20), there was a rebound in MSS CFP donor lifetime values of plasma membrane regions near mCherry-Gag signal that was recently departed compared to baseline MSS lifetime value (Fig 7B, bottom right panel, green line; S7D Fig)). In some cases, the drop in CFP average lifetimes (increased FRET) occurred with a few seconds delay after mCherry-Gag release (Fig 7B, bottom left panel). This drop, however, was recovered before the full release of Gag and right at the moment of fusion pore formation. The drop in local CFP lifetime is recovered before full Gag release (Fig 7B). These results indicate that during HIV-1$_{JR-FL}$ entry in reporter cells there is a local reduction in plasma membrane tension that is immediately restored following virus fusion.

When analysing MSS CFP donor lifetimes in 3 x 3 pixel plasma membrane regions overlapping with mCherry-Gag-containing viruses attempting to enter cells treated with 2-DG, we noticed that viruses unable to enter cells were associated with plasma membrane regions with relatively high membrane tension values when compared to vehicle conditions (Fig 7B, bottom right panel, blue line). These higher plasma membrane tension values were also present when analysing the whole cell (S7A Fig). Additionally, in 2-DG-treated conditions, we failed to find drops in MSS lifetime values in local PM regions or loss of mCherry-Gag signal (Fig 7B, bottom right panel) in all viruses analysed (n = 20), indicating that the localised nature of the plasma membrane tension drop during viral entry is inhibited by 2-DG. Remarkably, we recovered MSS lifetime profile traces similar to non-treated cells when 2-DG-treated cells were supplemented with water-soluble cholesterol (Fig 7B, bottom right panel, purple line). Altogether, our results illustrate that a cell's glycolytic activity sets surface cholesterol levels and plasma membrane tension which subsequently determines the success of HIV-1 fusion and thus ultimately HIV-1 infection.

## Discussion

Recent studies have stressed the impact of host metabolism on HIV infection [6,7,13]. It has been shown that differences in HIV-1 susceptibility between naive and more differentiated T cells were related to metabolic activity, in particular to oxidative phosphorylation and glycolysis [13]. Inhibition of glycolysis impaired HIV-1 infection in cell culture in all CD4+ T cell types. The mechanism behind these results and the link between metabolism and HIV-1 fusion, however has remained elusive until now. Our data with T cells utilising HIV-1 pseudoparticles corroborates the finding that acute inhibition of glycolysis inhibits productive infection (Fig 1). Moreover, we also show that inhibition of glycolysis specifically sequesters cholesterol from the plasma membrane in both T cells and TZM-bl cells (Fig 4). The combination of genetically encoded calibrated FRET-based biosensors detecting metabolic activity permitted the use of single-cell ATP/ADP and lactate as an indirect reporter for glycolytic activity (Figs 1 and 2). These experiments were combined with single cell fusion and infection assays and revealed an important finding: The metabolic landscape predetermined a cell's propensity for HIV-1 infection.

Various envelopes were utilised to pseudotype our HIV-1 particles: VSV-G, HXB2, NL4.3 and JR-FL. It is well-established that VSV-G pseudotyped virions enter host cells via

endocytosis [59]. However, the entry site of HIV-1 envelopes are debated and argued to be cell-type dependent [60]. It is thought that HIV-1 R5- and X4-tropic virions enter both primary T cells and T cell lines specifically at the plasma membrane, whereas endocytosis may be dispensable [61,62]. However, in reporter cells, it has been put forth that HIV-1 may productively enter host cells via endosomes [43]. Our report illustrates that in the absence of spinoculation HIV$_{JRFL}$ can fuse in both the plasma membrane and the endosomal compartments. Here, we focused on the events that do occur in the plasma membrane. Importantly, glycolytic activity and the regulation of physical properties in the plasma membrane via cholesterol content seem to be universally required regardless of the entry pathway. Indeed, we detected that this dependence of glycolytic activity and virus entry seemed to be relevant to all pseudotypes analysed (i.e. HIV-1$_{JR-FL}$, HIV-1$_{NL4.3}$, HIV-1$_{HXB2}$ and HIV-1$_{VSV-G}$).

In light of this, the membrane order and tension values obtained from the plasma membrane should be carefully interpreted when extending our discussion of these properties in the context of cholesterol content and virus entry in endosomes. The MSS tension probe was engineered to only label the plasma membrane and our analysis of FlipTR-stained cells was restricted to this region of the cell. Our analysis was restricted to the plasma membrane because FlipTR-labelled endosomes provided insufficient numbers of photons for reliable TCSPC FLIM analysis. Nevertheless, filipin-stained, TZM-bl cells labelled with LDL-Bodipy to identify early endosomes showed reduced cholesterol content in these compartments. These results reflect the fact that the majority of cellular cholesterol resides in the plasma membrane and reductions in plasma membrane cholesterol may be reflected downstream (i.e. in endosomes) [63]. Although further experiments are required to validate the subsequent changes in membrane order and tension upon endosomal cholesterol reduction during 2-DG treatment, given that we have seen reductions in endosomal cholesterol content and VSV-G fusion, it is possible that the membrane properties discussed in the plasma membrane which prevent plasma membrane entry of HIV-1$_{JR-FL}$ could be reflected in the endosomes.

Naïve, primary T cells are known to be exquisitely sensitive to metabolic perturbations, particularly mitochondrial-directed agents such as oligomycin, with even low levels substantially abrogating T cell proliferation and activation [64]. MT4 cells exist as transformed, pre-activated T cells with extraordinarily high glycolytic flux [65]. Similarly to previously reported studies utilizing pre-activated, transformed T cells [65], such as Jurkat and CEM cell lines, the treatment of MT4 cells with mitochondrial-directed agents (i.e. oligomycin) often fails to alter intracellular ATP pools, where treated cells remain energetically competent. Similarly, we illustrated that oligomycin treatment failed to not only alter the redox ratio as determined by two-photon microscopy (S2 Fig) but failed to alter the success of HIV-1 infection (Fig 1B). These findings, contextualised by prior studies, emphasize the importance of glycolytic flux and HIV-1 infection. Indeed, naïve T cells, heavily reliant on oxidative phosphorylation, are highly resistant to HIV-1 infection. However, differentiated T cell subsets are not only more less reliant oxidative phosphorylation as they become more glycolytically dependent, but are increasingly susceptible to HIV-1 infection [13,66]. Our study therefore adds to the discussion of how metabolism may influence HIV-1 infection at the level of entry. However, considering the fact our study solely scrutinized the role of glycolytic flux in the context of viral fusion, other metabolites which modulate immune function and differentiation (e.g. glutamine) should be explored. Indeed, metabolites such as glutamine are paramount for T cell differentiation and therefore likely play an important role for HIV-1 susceptibility.

Plasma membrane lipid bilayers contain a variety of lipids which are thought to be laterally segregated into domains to regulate fusion, endocytosis and signal transduction [67,68]. Lo domains are thought to be dominated by saturated lipids (i.e. sphingolipids) and sterols (i.e. cholesterol) surrounded by a pool of unsaturated phospholipids constituting Ld domains. The

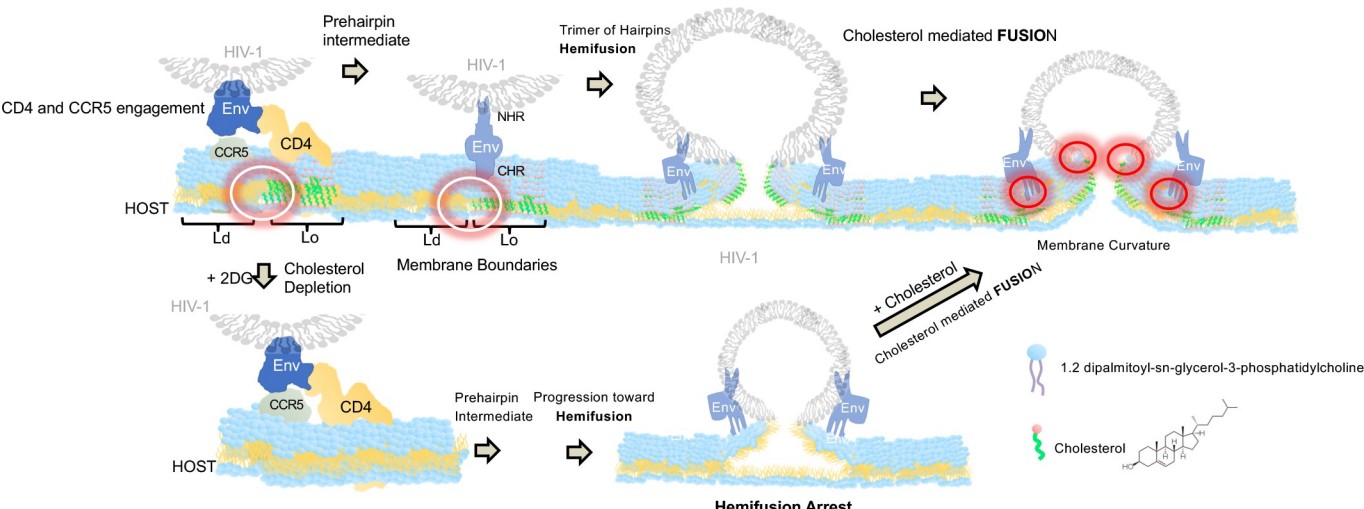

**Fig 8. Cholesterol availability regulates the transition between HIV-1 hemifusion and fusion.** HIV-1 Env sequentially interacts with CD4 and CCR5 at the boundaries of ordered lipid domains. The progression toward hemifusion is not cholesterol dependent as cells treated with 2DG had a reduced concentration of cholesterol and were able to progress toward HIV-1 hemifusion. Viruses exposed to cells pre-treated with 2DG were arrested right at hemifusion and only the addition of cholesterol rescued full fusion suggesting that cholesterol is a limiting factor in the HIV-1 fusion reaction in the process of fusion pore formation. White circles (left panels) represent lipid boundaries where cholesterol might induce HIV-1 Env priming and favour the pre-hairpin intermediate. Red circles (right panels) represent a potential role of cholesterol during fusion pore formation and membrane curvature.

contributions of cholesterol and the boundaries between Lo and Ld domains to the successful entry of viruses including HIV-1 are well-established. Membrane order and tension are significant contributors to several membrane-regulated processes in human cells important for viral infection, including viral entry (i.e. fusion and endocytosis) and egress [24,25]. Recently, several studies have offered a link between metabolic activity and sensing in live cells to cellular membrane tension indicating that various membrane-regulated processes are controlled by master metabolic sentinels [69–71]. Our results further build upon this link suggesting that high glycolytic activity may increase susceptibility of cells to support HIV-1 entry due to membrane tension regulation.

Specifically, our observation that decreased virus entry occurs following glycolytic inhibition could be linked to alterations in either membrane order or tension. We hypothesise that glycolysis directly, through the production of acetyl-CoA precursors, or indirectly regulates cholesterol in cellular membranes as the addition of 2-DG inhibited cholesterol accumulation without affecting CD4 or co-receptor expression levels (S4 Fig). This loss in cholesterol led to drastic increases in membrane tension and loss of membrane order, owing to the sequestration of cholesterol from the plasma membrane (Fig 5 and S7 Fig). These two aspects are crucial for the HIV-1 fusion reaction for many reasons. First, it has been shown that lipid-ordered (i.e. cholesterol-laden) and lipid-disordered membrane domains (also termed raft boundaries) can act as an attractor for HIV Env [29,72]. Both experimental and theoretical studies show that the presence of raft boundaries in the vicinity of HIV-1 entry sites reduces the energy barrier required to perform virus-cell fusion [21]. Moreover, the line tension created at lipid domain boundaries drives gp41-mediated fusion, owing to the mismatches in lipid length occurring at Lo and Ld boundaries, most likely creating membrane curvature which favours virion fusion [21] (Fig 8). Cholesterol and other ordered lipids are known to maintain negative spontaneous curved membrane domains, which facilitates the creation of a fusion stalk by reducing its formation energy in addition to enabling its progression to fusion pore formation [73]. The sequestration of cholesterol from the host membrane in glycolytically-inactivated cells arrests

HIV fusion, as this line-tension necessary to drive fusion is diminished. Furthermore, sequestration of cholesterol from the host plasma membrane ostensibly increases membrane tension (Fig 8), which has been illustrated previously to collapse hemifusion intermediates [74] in addition to preventing the formation of highly curved intermediates, disfavouring packing defect formation and the generation of nucleation points for two membranes to be joined [73]. Moreover, this increase in membrane tension might in turn have an effect on the optimal hydrophobic equilibrium between ordered and disordered lipid domains needed to complete the fusion reaction [21,29].

It is quite possible that glycolytic activity may be related to the availability of other lipids on the plasma membrane that could be essential for HIV-1 entry, in addition to other viruses. Indeed, phospholipid biosynthesis is linked to glycolysis via dihydroxyacetone phosphate, formed from glycerol-3-phosphate via its reduction by NADH. Furthermore, it is known that certain phospholipids act to enhance HIV-1 entry, and that chemically-modified phospholipids alter the curvature of the membrane, which can modify virus entry [75]. Future experiments should investigate whether glycolytic activity alters the availability of other phospholipids known to enhance HIV-1 entry.

We hypothesise that there is a range of hydrophobic mismatch between ordered and disordered lipid domains that renders HIV fusion plausible. When this equilibrium is lost (both toward high or low tension) the energy required for gp41 fusion peptide to complete fusion (expressed in tens of kBT, [21]) is not compatible with the completion of the fusion reaction. This is why depletion of cholesterol might allow HIV Env to be primed with CD4 and CCR5 followed by arrest at hemifusion (Fig 3) even if overall tension in the host cell membrane is increased (Fig 8). Moreover, our data suggests that cholesterol is a limiting factor for the transition between HIV-1 hemifusion and fusion (Fig 8) and not necessarily during the sequential interactions between HIV-1 Env with CD4 and CCR5 co-receptor at phase boundaries [29]. Indeed, in cells treated with 2DG where cholesterol is sequestered, HIV-1 was able to progress until hemifusion. The external addition of cholesterol allowed full fusion to be completed (Fig 7).

Our results indicate that glycolytic activity is needed to maintain higher lipid order and lower levels of membrane tension inside cells, two characteristics that promote HIV-1 fusion with cells. Furthermore, we have also shown that HIV-1 induces broad fluctuations in host global membrane tension during virus entry, yet require localised, transient reductions in plasma membrane tension in order to enter their host cell. Moreover, our results with primary cells (S3 and S6 Figs) reinforce our claims that host cell glycolysis regulates cholesterol availability and HIV-1 entry. In brief, we demonstrated that treatment of primary CD4+ T cells with 2-DG led to similar decreases in HIV-1$_{HXB2}$ entry and that treatment with cholesterol-lowering medication (i.e. simvastatin) recapitulated this result (S3 Fig). Furthermore, 2-DG treatment led to similar cell surface cholesterol reductions (S6 Fig). Ultimately, our work shows that glycolytic inhibition with 2-DG creates changes in global and local membrane tension that restricts HIV-1 fusion at the hemifusion stage and that this inhibition can be tied to cholesterol availability.

## Material and methods

### Cell culture

TZM-bl (already existing collection from Quentin Sattentau lab) and Lenti-X-293T (Takara Bio, Europe, France) cells were cultured in either complete Dulbecco's Modified Eagle Medium (DMEM) or DMEM F-12 (Life Technologies), respectively, both of which contained 10% fetal bovine serum (FBS), 1% Penicillin-Streptomycin (PS) and 1% L-glutamine

(LGlut). Cells were maintained at 37˚C which provided 5% $CO_2$. MT4 cells (already existing collection from Alex Compton lab) were cultured in RPMI containing 10% FBS, 1% PS and 1% L-Glut.

CD4 T cell lymphoblasts were generated from human peripheral blood CD4 T cells isolated from healthy donors (Levine et al., 1997). Briefly, CD4 T cells were isolated by negative selection (RosetteSep Human CD4$^+$ T cell Enrichment Kit, StemCell technologies) following the manufacturer's procedure. When referring to rested CD4 T cells, these T cells are kept overnight in complete medium (RPMI 1640 media supplemented with 10% heat-inactivated fetal bovine serum, 50 U/ml of Penicillin-Streptomycin, 2 mM L-Glutamine, 10 mM HEPES, 1 mM Sodium Pyruvate, and 100 µM non-essential amino acids) prior to experimental use. In any other case, CD4 T cells were activated for 2 days using anti-CD3/anti-CD28 T-cell activation and expansion beads (Dynabeads, ThermoFisher Scientific) in complete medium supplemented with 100 U/ml of recombinant human IL-2 (PeproTech). After 2 days, T cells are separated from beads and incubated for 3 days in fresh IL2-containing medium at a concentration of $1 \times 10^6$ cells/ml prior to experimental use.

## Reagents and antibodies

All chemical and biochemical reagents were obtained from the following sources: The Flipper-TR probe was purchased from Spirochrome (Geneva, Switzerland). Antibodies against CD4 conjugated to PercP Cy5.5 (ab161a1) and CCR5 conjugated to FITC (ab11466) were from Bio-Legend (San Diego, CA, USA) and Abcam (Cambridge, UK), respectively. β-lactamase CCF2-AM loading solutions from the LiveBLAzer FRET B/G Loading Kit was obtained from Life Technologies (Carlsbad, CA, USA); Filipin, 2-deoxy-glucose (2-DG), Methyl-β-cyclodextrin (MBCD) and water-soluble cholesterol [76] was obtained from Sigma-Aldrich (St. Louis, MO, USA). Low density lipoprotein (LDL) from human plasma (BODIPY) complex and Simvastatin were obtained from ThermoFisher.

## Plasmids transfections

The pR8ΔEnv plasmid (encoding HIV-1 genome harbouring a deletion within Env), pcRev, VPR-BlaM, Gag-eGFP and VSV-G were kindly provided by Greg Melikyan (Emory University). The plasmid encoding the JR-FL envelope protein was a kind gift from James Binley (Torrey Pines Institute for Molecular Studies). The MSS sensor was kindly provided from Bo Liu (Dalian University of Technology). Perceval and Laconic plasmids were obtained from Addgene. Transient transfections of Perceval, Laconic (Addgene plasmid numbers 21737 and 44238, respectively) and the MSS probe was performed according to the manufacturer's protocol for GeneJuice (Novagen). Transiently-transfected cells were analysed by FLIM 24–48 hours later.

## Virus production

Vpr-BlaM-containing, JR-FL and VSV-G pseudotyped viral particles were produced via transfection of 60–70% confluent Lenti-X-HEK-293T cells seeded in T175 flasks. Component DNA plasmids were transfected into Lenti-X-HEK-293T cells via GeneJuice (Novagen) according to manufacturer's protocol. Specifically, cells were transfected with 2 µg pR8ΔEnv, 1 µg pcREV, 3 µg of the appropriate viral envelope (either VSV-G, the CCR5-tropic HIV-1 strain JR-FL, or the CXCR4-tropic HIV-1 strain NL4.3), and either 2 µg Vpr-BlaM if to be used for β-lactamase assays, or 3µg of eGFP-GagΔEnv if being used for flow cytometry experiments. Alternatively, if to be utilised for single particle tracking, 2 µg pR8ΔEnv, 3 µg Gag-eGFP-Δenv, 1 µg pcREV and 3 µg of the appropriate viral envelope were transfected into HEK-293T cells. If to be

utilised for single particle tracking whilst recording MSS FLIM lifetime acquisition, 2 μg pR8ΔEnv, 2 μg mCherry-2xCL-YFP-Vpr, 1 μg pcREV and 3 μg of the appropriate viral envelope. All transfection mixtures were then added to cells supplemented in complete DMEM F12, upon which time they were incubated in a 37 ˚C, 5% $CO_2$ incubator. 12 hours post-transfection, the medium was replaced with fresh, phenol-red free, complete DMEM F12 after washing with PBS. Cells were subsequently incubated for a further 24 hrs. If viruses were to be utilised for single particle tracking, 12 hours post-transfection, the transfection mixture was removed and cells were washed twice with PBS before being incubated at 37º C with DMEM F12 medium containing 10μM DiD (Life Technologies) for 4 hour. The staining mixture was subsequently removed, upon which time the cells were washed twice with PBS and DMEM F12 was then added again and incubated for an additional 24 hours prior to harvesting as follows. 48 hrs post-transfection for either Vpr-BlaM, double-labelled or immature virions, the supernatant containing virus particles was harvested, filtered with a 0.45 μm syringe filter (Sartorius Stedim Biotech), aliquoted and stored at −80 ˚C.

## Virus titering

To titer the produced pseudovirions, 2 x $10^4$ TZM-bl cells were seeded in complete DMEM, in triplicate, in a 96-well plate (Falcon) and allowed to grow in a 37 ˚C, 5% $CO_2$ incubator for 24 hours. Subsequently, the media was removed and replaced with 10-fold serially-dilutions of the produced pseudovirus particles. Aliquoted Vpr-BlaM-containing or double-labelled JR-FL and VSV-G pseudotyped viral particles additionally contained the β-galactosidase gene in the pr8ΔEnv construct. Cells were incubated for an additional 48 hours in a 37 ˚C, 5% $CO_2$ incubator, upon which time the medium was removed, the cells were then washed with PBS and fixed with 2% paraformaldehyde for 10 minutes. After an additional PBS wash post-fixation, an X-gal (i.e. BCIG, or 5-bromo-4-chloro-3-indolyl-β-D-galactopyranoside) solution comprised of 500mM $K_3[Fe(CN)_6]$, 250mM $K_4[Fe(CN)_6]$, 1M $MgCl_2$, PBS, and 50mg/mL Xgal was added with to the cells and incubated for two hours at 37 ˚C, 5% $CO_2$ incubator in the dark. Afterwards, cells were washed with PBS and subsequently imaged with a 630nm excitation continuous laser (Leica, Manheim) while recording the emission spectra with a 650-750nm emission window, pixel by pixel (512 x 512) using a Leica SP8 X-SMD microscope with a lambda resolution of 12 nm, in order to count infection-positive cells for each 10-fold dilution to extrapolate the virus titer.

## Single virus tracking: Spun vs non-spun cells

HIV pseudoviruses decorated with JRFL Env and labelled with DiD in the membrane and Gag-GFP were exposed on the top of TZM-bl cells at 4˚C. These cells were either spun for 30 min at 2100 G in a refrigerated centrifuge, or left 30 min in the cold so that the HIV particles would sediment down on the surface of the TZM-bl cells. In both cases the particles that were pullulating around and not primed were washed out with cold PBS. The observation chambers were then put under the microscope set at 37˚C and the cold medium immediately replaced by warm medium set at 37˚C right at the moment when we started the HIV fusion reaction and the imaging acquisition procedure. We employed the SP8 X SMD confocal microscope described also below. Briefly a white laser (Leica, Germany) and an Argon laser were combined (633nm and 488nm) employing a 63X 1.4 NA objective to excite DiD and Gag-GFP respectively alternating lines to avoid emission bleed-through. The emission photons for both fluorescent signals were detected using HyD photon counting detectors. A third detector (PMT) was also used to recover the transmission light simultaneously.

### β-lactamase assay

Four-hours post-addition of the pseudotyped virus particles harbouring Vpr-BlaM at the specified MOI to the transfected TZM-bl cells, virus particles were removed, and cells were washed twice with PBS. After the washes, complete DMEM was added to each well and the cells were allowed to incubate at 37°C, 5% $CO_2$. Cells were then loaded with CCF2-AM from the Live-BLAzer FRET—B/G Loading Kit (Life Technologies) and incubated at room temperature in the dark for 2 hrs. After this incubation step, the CCF2 was removed, and cells were washed with PBS and maintained in 2% FBS-containing, phenol-red free DMEM prior to imaging.

### β-lactamase assay spectral analysis

CCF2-loaded TZM-bl or primary rested CD4+ T cells were excited with a 405nm continuous laser (Leica, Manheim) while recording the emission spectra with a 430-560nm window, pixel by pixel (512 x 512) using a Leica SP8 X-SMD microscope with a lambda resolution of 12 nm. Utilising ImageJ (http://imagej.nih.gov/ij/), the ratio of the blue (440-480nm, cleaved CCF2) to green (500-540nm, uncleaved CCF2) emission was calculated pixel-by-pixel using 20X objective. Statistical analyses of the BlaM data were performed using a two-tailed Fishers Exact Test (SigmaPlot, San Jose, CA). Cells were then removed of the CCF2 mixture and allowed to incubate for 24 hours in complete DMEM if they were to be analysed for infection by the B-Gal assay (below) for infection.

### β-galactosidase assay

After imaging the CCF2-loaded TZM-bl cells, cells were washed with PBS and maintained in 2% FBS-containing, phenol-red free DMEM overnight for 12 hours in a 37 °C, 5% $CO_2$ incubator. At least 36 hours after the addition of the pseudotyped virus particles, the medium was removed, and the cells were then washed with PBS and subsequently fixed with 2% paraformaldehyde for 10 minutes. After an additional PBS wash post-fixation, an X-gal (i.e. BCIG, or 5-bromo-4-chloro-3-indolyl-β-D-galactopyranoside) solution comprised of 500mM $K_3[Fe(CN)_6]$, 250mM $K_4[Fe(CN)_6]$, 1M $MgCl_2$, PBS, and 50mg/mL Xgal was added to the cells, upon which time the cells were incubated for two hours in a 37 °C, 5% $CO_2$ incubator in the dark. Afterwards, cells were washed with PBS and subsequently imaged with a 630nm excitation continuous laser (Leica, Manheim) while recording the emission spectra with a 650-750nm emission window, pixel by pixel (512 x 512) using a Leica SP8 X-SMD microscope with a lambda resolution of 12 nm, in order to identify infection-positive cells.

### Filipin immunofluorescence staining

TZM-bl cells, MT4 cells or PBMCs were treated with either a vehicle control, MBCD, water-soluble cholesterol or various concentrations of 2-DG for 2 hours were fixed with 4% PFA for 15 minutes at room temperature on Ibidi 35mm gridded dishes. The fixative was washed away three times with cold PBS and then quenched with 1.5mg/mL glycine for 10 minutes at room temperature. Cells were then stained with filipin at 50μg/mL for 2 hours at room temperature, protected from light. The filipin stain was washed away three times with PBS and then cells were subsequently imaged using a SP8–X-SMD microscope with the FALCON module from Leica Microsystems (Manheim, Germany). If cells were to be analysed for endosomal filipin content, cells were loaded with LDL-Bodipy at a concentration of 10 μg/mL and allowed to incubate at 37 °C, 5% $CO_2$ for 2 hours. The LDL-Bodipy complex was then removed, cells were washed three times with room temperature DPBS. Following washing, cells were then treated with specified treatment conditions as described (e.g. vehicle or 2-DG treatment).

After treatment, cells were fixed and stained with filipin and imaged as described above. Z-stacks were acquired with a step-size of 0.3 μm for colocalization studies of the median 5–10 slices in each stack for each image of each condition.

## Fluorescence lifetime imaging microscopy (FLIM)

Live TZM-bl cells expressing Perceval, Laconic or FlipTR were imaged using a SP8–X-SMD Leica microscope, whilst MSS-expressing cells were imaged using a SP8–X-SMD microscope with the FALCON module, both from Leica Microsystems (Manheim, Germany). Cells of interest were selected under either a 20x air-immersion or 63x oil-immersion objective (i.e. Perceval, Laconic and FlipTR) or a 100x/1.4 oil immersion objective (i.e. MSS). Perceval-expressing cells were excited using a 488 nm pulsed laser, Laconic-expressing cells were excited using a 470 nm pulsed laser, both tuned at 40 MHz coupled with single photon counting electronics (PicoHarp 300) and subsequently detected by hybrid external detectors in photon counting mode. FlipTR stained cells were excited with a 488 nm puled laser tuned at 20MHz.

MSS-expressing cells were subjected to a 440nm picosecond pulsed diode laser PDL 800-B (PicoQuant) tuned at 40MHz in order to excite the FRET donor (CFP) while the emitted photons passing through the 450-480nm emission filter were detected using the internal hybrid detector in photon counting mode. Time-domain FLIM experiments were performed using a TCSPC approach operated by the FALCON module (Leica Microsystems, Manheim, Germany) integrated within the Leica SP8-X-SMD microscope.

In order to remove artefacts caused by noise or photo–bleaching and insufficient signal to noise, cells with negligible amounts of bleaching and at least 250–1000 photons per pixel were only allowed in the analysis. Symphotime 64 software (Picoquant) was utilized to acquire the fluorescence decay of each pixel in individual cells expressing Perceval, Laconic, or Flipper-TR-stained cells, which was deconvoluted with the instrument response function (IRF) and fitted by a Marquandt nonlinear least–square algorithm with two–exponential models. The mean fluorescence lifetime ($\tau$) was calculated as previously described using Symphotime. Statistical analysis of the lifetime data was performed using a two-tailed t-test (Origin, Northhampton, MA, USA).

## Label free NADP(H) two photon FLIM

A two photon FLIM SP8 DEEP DIVE FALCON microscope (Leica Microsystems, Mannheim, Germany) equipped with 3 non-descanned photon counting detectors was employed to measure the autofluorescence of NAD(P)H in live cells (both MT4 T cells and TZM-bl cells) as explained in [77]. Briefly, the two-photon laser was tuned at 730 nm and the emission window was set at 450–500 nm. A 100X 1.4 NA corrected for IR was employed to recover all images. Images were collected so that at least 100–1000 photons per pixel were acquired after 3–5 min. A two exponential analysis fitting pixel-by-pixel the fluorescence decay from all pixels in all images was performed (FALCON Leica software, Germany) to recover the ratio of protonated and non-protonated NAD(P)H. Regions of interest comprising individual cells treated with and without 2-DG gave different ratios as shown in S2 Fig. Phasor plots [78]were employed to show the shift in NADP(H) concentration when comparing cells (both TZM-bl and MT4) in either vehicle or 2-DG-treated conditions. Leica software (LASX) was employed to produce the phasor plots (Leica Microsystems, Mannheim, Germany).

## Flow cytometry

Single-round infections were performed either with HIV-1$_{VSV-G}$-eGFP-Gag or HIV-1$_{NL4.3}$-eGFP-Gag. MT4 T cells were infected in triplicate (1 x 10$^5$ cells/ well, 200uL) at an MOI of 1.

Active HIV-1 infection was estimated by flow cytometry (BD LSRII, BD bioscience) as the percentage of eGFP-expressing, live MT4 T cells 48 hours after infection, and live cells were determined via LIVE/DEAD Fixable Near-IR Dead cell stain for 633/635 nm to stain dead cells following manufacturer's instructions (Invitrogen).

## Extrapolating the ATP:ADP ratio and relative lactate concentrations and MSS lifetimes

Utilizing Symphotime software (Picoquant Gmbh, Berlin, Germany), each whole cell expressing Perceval or Laconic was determined to be a region of interest. The region of interest with the smallest lifetime ($\tau_{int}$) was selected as the cell with the highest amount of ATP relative to ADP in the field of view analysed. From this region of interest, the short lifetime component ($\tau_1$) was fixed in all other analysed cells when acquiring the TCSPC fluorescence decay of each pixel, as this component possessed the highest amplitude $a_1$, therefore representing the relative ATP concentration in the cell. The lifetime component $\tau_2$ was interpreted to represent the relative contribution from ADP binding to the Perceval sensor, and therefore its corresponding amplitude $a_2$ was utilized to extract the relative ADP concentration. The resulting two-exponential decay, which was deconvoluted with the instrument response function (IRF) and fitted by a Marquandt nonlinear least–square algorithm contained two amplitudes, $a_1$ and $a_2$. The ratio (i.e. $a_1/a_2$) of these two amplitudes was taken to generate the ATP:ADP ratio. To extract the relative lactate concentration from Laconic, a similar approach was utilised, where the amplitude $a_1$ of the long lifetime component $\tau_1$ was divided by the sum of the amplitudes from the two-component exponential decay.

For the MSS probe, we utilised the internal FALCON module Leica Microsystems, Manheim, Germany) integrated within the Leica SP8-X-SMD microscope for our FLIM analysis. Whole cells expressing the MSS probe or 3 x 3 pixel-by-pixel regions of the MSS-labelled membrane overlapping with mCherry-labelled virions were chosen as our region of interest. To increase photon counts per pixel, we binned 5 x 5 pixels in addition to grouping frames of our acquisition in groups of 10. We utilised a three-exponential decay deconvoluted with the IRF and fitted by a Marquandt nonlinear least-square algorithm containing three amplitudes with the third long lifetime component fixed to the average of all cells analysed to isolate short $\tau_1$ and long $\tau_2$. The longer lifetime component ($\tau_2$) was chosen to represent the donor lifetime and therefore the membrane tension. When analysing 3 x 3 pixel-by-pixel regions of the MSS-labelled membrane overlapping with mCherry-labelled virions, to accommodate for low photon counting numbers, we utilised a single-exponential decay to extract MSS $\tau$.

## Statistics

All statistical calculations (t-test, ANOVA, standard deviation and error) were calculated using Originlab software (Northhampton, USA)

## Supporting information

**S1 Fig. Acute 2-deoxyglucose treatment abrogates HIV-1$_{NL4.3}$ infection in MT4 cells and induces fluorescent lifetime changes in biosensors Laconic and Perceval.** A.). Bar charts depicting % eGFP-expressing cells as a marker of infection illustrating that acute treatment with increasing concentrations of 2-DG led to reductions in HIV-1$_{NL4.3}$ infection in human MT4 T cells. B.) Representative intensity (left) and fluorescent lifetime imaging (right) of single cells transiently expressing intracellular lactate biosensor Laconic in single TZM-bl cells with increasing concentrations of glycolytic inhibitor 2-DG; scale bar, 50μm. C.) Representative intensity (left) and fluorescent lifetime imaging (right) of single cells transiently expressing

intracellular ATP:ADP ratio biosensor Perceval in single TZM-bl cells with increasing concentrations of glycolytic inhibitor 2-DG; scale bar, 50μm. D.) Dot plots representing lifetimes of intracellular lactate biosensor Laconic extracted from live, single cells as regions of interest post-treatment for 2-hours with increasing concentrations of 2-DG in TZM-bl cells. E.) Dot plots representing lifetimes of intracellular ATP:ADP ratio biosensor Perceval extracted from live, single cells as regions of interest post-treatment for 2 hours with increasing concentrations of 2-DG in TZM-bl cells.
(TIF)

**S2 Fig. Acute treatment of 2-deoxyglucose, not oligomycin, inhibits glycolytic flux in a pH-independent manner cell lines.** A.) Representative images (left) and phasor plots (right) representative of FLIM distributions of NAD(P)H alone (top row), vehicle-treated conditions in TZM-bl (left) and MT4 cells (right) (middle row) and acute treatment with 2-DG (bottom row); scale bar 5 μm. Phasor FLIM plots illustrate each pixel converted via Fourier Transform to the phase domain. The phasor plots illustrate longer lifetimes (i.e. enzyme-bound NAD(P)H, lower glycolytic flux) to the left and shorter lifetimes (free NAD(P)H, higher glycolytic flux) to the right. B.) Bar charts representing the lifetime extracted from single TZM-bl cells expressing intracellular pH biosensor pHRed indicating a lack of change in fluorescence lifetime during acute 2-DG treatment. C.) Box plot representing the ratio of $NAD(P)H_{free}$ vs. $NAD(P)H_{protein-bound}$ in MT4 cells treated with oligomycin. D.) Box plot representing the ratio of $NAD(P)H_{free}$ vs. $NAD(P)H_{protein-bound}$ in MT4 cells treated with 2-DG. The presented two-photon FLIM data was acquired as described in material and methods. Box plots represent data acquired from at least 30 cells per condition acquired from three independent experiments.*** p<0.001 as determined by one-way student's T-test.
(TIF)

**S3 Fig. Acute treatment with 2-DG or simvastatin abrogates HIV-1$_{HXB2}$ fusion in primary CD4+ T cells.** A) Primary cells were exposed to either naked (i.e. No Env) HIV-1 or HIV-1$_{HXB2}$ virions and treated with vehicle, 100 mM 2-DG or 10μM Simvastatin. Brightfield images (first row) show that in all cases the integrity of the cells was maintained. The BlaM assay for HIV fusion (Blue/Green channel ratio images, second row) shows that the number of positive fusion cells (red) is higher for cells only exposed to HIV$_{HXB2}$. Cells exposed to both HIV$_{HXB2}$ and 100 mM 2-DG or 10uM Simvastatin were less susceptible to HIV$_{HXB2}$ fusion as limited fusion positive cells (red cells) were detected. The pixel-by-pixel histograms for each condition are also shown for each condition in the lowest row. B) When quantifying the overall populations of cells (i.e. at least100 cells per condition) and taking as a negative control No Env HIV$_{HXB2}$ (gray dots, and straight gray line) as a threshold for fusion, one could see that in the green channel versus blue channel plots (based in average intensities recorded from single cells) 10.1% of primary T cells were fusion positive when exposed to HIV$_{HXB2}$ (red dots above the grey No Env threshold line in the left panel). For cells treated with 100 mM 2-DG, only 2.2% turned out to be fusion positive (red dots above the grey line in the middle panel). In turn, for cells treated with 10μM Simvastatin, only 2.4% were fusion positive (green dots above the grey No Env threshold line, right panel).
(TIF)

**S4 Fig. Acute treatment with 2-DG does not alter cell viability or cell-surface receptor expression, and single virus tracking of HIV-1$_{JR-FL}$ in vehicle or 2-DG-treated conditions.** A.) Bar charts depicting the percentage of dead TZM-bl cells detected by propidium iodide (PI) staining in single cells treated with increasing concentrations of 2-DG for two hours. B.) Bar charts representing normalized HIV-1$_{JR-FL}$ fusion relative to vehicle in single TZM-bl cells

as determined by the β-lactamase assay in cells treated with glucose-free medium for two hours before virus addition. C.) Bar charts illustrating that relative CD4 and CCR5 expression levels do not drastically change during listed treatment conditions. Bar charts shown in the panel are representative of a mean of three independent experiments. D.) Representative fluorescence series of images (left) and single particle tracking traces (right) of Gag-eGFP (green) and DiD (red) dual-label HIV-1$_{JR-FL}$ pseudovirions in TZM-bl cells (scale bar) illustrating that in control conditions (i.e. no 2-DG) that HIV-1$_{JR-FL}$ entry in TZM-bl cells proceeds with a precipitous loss of Gag-eGFP and maintenance of DiD signal, indicative of endocytosis as previously described (n = 15, acquired during three independent experiments). E.) Representative fluorescence series of images (left) and single particle tracking traces (right) of Gag-eGFP (green) and DiD (red) dual-label HIV-1$_{JR-FL}$ pseudovirions in TZM-bl cells (scale bar) illustrating that in 2-DG-treated conditions (i.e. 100mM 2-DG) that HIV-1$_{JR-FL}$ entry in TZM-bl cells may also proceed with a slow decay of DiD signal and maintenance of eGFP-Gag signal, indicative of hemifusion (n = 15, acquired during three independent experiments). $^*$p<0.5, $^{**}$p<0.01 $^{***}$ p<0.001 as determined by either one-way ANOVA or one-way student's T-test. (TIF)

**S5 Fig. Single virus tracking of HIV-1$_{JR-FL}$ viruses comparing spun vs. non-spun TZM-bl cells.** A.) HIVJR-FL pseudoviruses labelled with DiD and Gag-GFP exposed on TZM-bl at 4˚C during 30 min as described in material and methods were placed under the microscope at 37˚C to start the entry process. White dotted circles indicate the HIV-1 particles co-labelled with both colours (red for DiD and green for Gag-GFP). A subset of these particles are only labelled with either Gag-GFP (green dots) or DiD (red dots). These particles were ignored. Scale Bar = 10 μm B.) Three regions of interest showing: i) endosomal fusion (first row, where a yellow particle turns red signifying DiD redistribution within the endosomal compartment and Gag-GFP release); ii) plasma membrane (PM) fusion (second row, where a double labelled particle designated with an arrow first changes from yellow to green indicating hemifusion (i.e. DiD dilution on the PM) and then Gag-GFP disappearance. A white dotted particle undergoing endosomal fusion is also shown indicating that the focus was kept stable during the acquisition and also that both phenomena can occur in non-spinoculated cells simultaneously. The third row shows a particle undergoing hemifusion (unable to progress toward full fusion as the GFP-Gag remains through time). Scale bar = 1 μm C.) Left panel: The statistics for spun (dark green) versus non-spun (light green) cells are shown. The proportion of events (0 to 1) for endosomal fusion, hemifusion, colour separation (capsid release) and plasma membrane fusion are shown for a total of n = 137 and n = 56 double labelled particles for spun and non-spun TZM-bl cells respectively. Also, a small fraction of events for both spun and not-spun conditions showed colour separation; colour separation is defined as double-labelled (yellow viruses) internalized within endosomes (labeled with DiD and Gag-GFP) undergoing fusion. Right at the moment of fusion DiD redistributes around the endosomal membrane and partially uncleaved Gag-GFP is released into the cytosol.The right panel depicts the cumulative distribution kinetics for HIV-1 particles undergoing: i) Fusion for spun cells (solid green dots); ii) hemifusion for non-spun cells (solid red dots), iii) full endosomal fusion for spun cells (open red dots) and iv) plasma membrane fusion non-spun cells (open green dots). (TIF)

**S6 Fig. Acute treatment with 2-DG leads to cell surface cholesterol reduction in cell lines and primary cells in addition to the endosomal compartment.** A.) Bar charts depicting normalized filipin mean fluorescence intensity per cell in various treatment conditions listed to illustrate that 2-DG treated cells have plasma membrane cholesterol content similar to 1mM

MBCD-treated cells and that supplementation with increasing concentrations of water-soluble cholesterol in 2-DG treated conditions rescues filipin staining signal. B.) Bar charts depicting filipin mean fluorescence intensity per MT4 T cell normalised to vehicle, indicating two-hour incubation of increasing concentrations of oligomycin do not alter surface plasma membrane cholesterol. C.) TZM-bl cells were grown in vehicle (top) or 2-DG-treated (bottom) conditions for two hours after being loaded with LDL-Bodipy staining to label the endosomal compartment. Filipin staining was used to label cholesterol at both the cell surface (red) and LDL-Bodipy (blue) identified regions (green illustrates the LDL-Bodipy associated cholesterol identified by LDL-Bodipy and filipin co-labelling); scale bar 10μm. D.) Bar charts illustrating normalized filipin mean fluorescence intensity in LDL-Bodipy labelled compartments in vehicle and 2-DG-treated conditions (top) and colocalization coefficients of LDL-Bodipy with filipin quantified by z-stacks (bottom). E.) (Left) Bar charts depicting raw filipin mean fluorescence intensity per primary CD4+ T cell in each condition listed. (Right) Representative images of primary CD4+ T cells stained with filipin after treatment conditions listed. Mean values were calculated from at least 50 cells per condition from three independent experiments. $^{*}p<0.5$, $^{**}p<0.01$ $^{***}$ $p<0.001$ as determined by either one-way ANOVA or one-way student's T-test.
(TIF)

**S7 Fig. Multiplexed FLIM with SVT reveals no drop in local tension during single HIV-1 fusion in live cells treated with 2-DG.** A.) Compiled MSS lifetimes extracted from whole TZM-bl cells compiled from acquisitions of multiple viral fusion events (green) compared to 2-DG treated cells where viruses were incapable of fusion (blue) or rescued events due to cholesterol treatment (purple). Each point represents a mean of at least 20 cells obtained during three separate experiments. B.) Bar charts comparing the MSS lifetimes of localised regions overlapping with mCherry-labelled HIV-1$_{JR-FL}$ during virion fusion (i.e. mCherry signal loss) with virus-free control areas in the same cell at the timepoint of fusion in TZM-bl cells. Each condition represents a mean of at least 20 cells obtained during three separate experiments $^{*}p<0.5$, $^{**}p<0.01$ $^{***}$ $p<0.001$ as determined by either one-way student's T-test C.) Bar charts comparing the MSS lifetimes of localised regions overlapping with mCherry-labelled HIV-1$_{JR-FL}$ (i.e. mCherry signal) with virus-free control areas in the same cell at the timepoint before fusion in TZM-bl cells. Each condition represents a mean of at least 20 cells obtained during three separate experiments $^{*}p<0.5$, $^{**}p<0.01$ $^{***}$ $p<0.001$ as determined by either one-way student's T-test. D.) Bar charts comparing the MSS lifetimes of localised regions overlapping with regions where mCherry-labelled HIV-1$_{JR-FL}$ fused with the host cell (i.e. mCherry signal lost) with virus-free control areas in the same cell at the timepoint after fusion in TZM-bl cells. Each condition represents a mean of at least 20 cells obtained during three separate experiments $^{*}p<0.5$, $^{**}p<0.01$ $^{***}$ $p<0.001$ as determined by either one-way student's T-test.
(TIF)

## Acknowledgments

We thank Luis Alvarez for technical support on advanced microscopy, acquisition, and analysis. We thank Bo Liu for sharing the KMSS and MSS biosensors.

## Author Contributions

**Conceptualization:** Alex A. Compton, Sergi Padilla-Parra.

**Data curation:** Charles A. Coomer, Sergi Padilla-Parra.

**Formal analysis:** Charles A. Coomer, Irene Carlon-Andres.

**Funding acquisition:** Michael L. Dustin, Alex A. Compton, Sergi Padilla-Parra.

**Investigation:** Charles A. Coomer, Irene Carlon-Andres, Maro Iliopoulou, Ewoud B. Compeer, Sergi Padilla-Parra.

**Methodology:** Charles A. Coomer, Irene Carlon-Andres, Ewoud B. Compeer, Alex A. Compton, Sergi Padilla-Parra.

**Project administration:** Michael L. Dustin, Alex A. Compton, Sergi Padilla-Parra.

**Resources:** Michael L. Dustin, Sergi Padilla-Parra.

**Software:** Charles A. Coomer, Irene Carlon-Andres, Sergi Padilla-Parra.

**Supervision:** Michael L. Dustin, Ewoud B. Compeer, Alex A. Compton, Sergi Padilla-Parra.

**Validation:** Charles A. Coomer, Irene Carlon-Andres.

**Visualization:** Charles A. Coomer, Sergi Padilla-Parra.

**Writing – original draft:** Charles A. Coomer, Sergi Padilla-Parra.

**Writing – review & editing:** Charles A. Coomer, Irene Carlon-Andres, Maro Iliopoulou, Michael L. Dustin, Ewoud B. Compeer, Alex A. Compton, Sergi Padilla-Parra.

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
