## [Decision Letter · Decision Letter 0]

21 Nov 2019

Dear Dr Padilla-Parra,

Thank you very much for submitting your manuscript "Single-cell glycolytic activity regulates membrane tension and HIV-1 fusion" (PPATHOGENS-D-19-01762) for review by PLOS Pathogens. Your manuscript was fully evaluated at the editorial level and by independent peer reviewers. The reviewers appreciated the attention to an important problem, but raised some substantial concerns about the manuscript as it currently stands. These issues must be addressed before we would be willing to consider a revised version of your study. We cannot, of course, promise publication at that time.

We therefore ask you to modify the manuscript according to the review recommendations before we can consider your manuscript for acceptance. Your revisions should address the specific points made by each reviewer.  The use of the most biologically relevant cells always strengthens a body of work so I am sensitive to the concern that most of the work is done with TZM-bl cells.  Also, it is disconcerting that the reviewers had such a hard time following the figures and data. 

(1) A letter containing a detailed list of your responses to the review comments and a description of the changes you have made in the manuscript. Please note while forming your response, if your article is accepted, you may have the opportunity to make the peer review history publicly available. The record will include editor decision letters (with reviews) and your responses to reviewer comments. If eligible, we will contact you to opt in or out.

(2) Two versions of the manuscript: one with either highlights or tracked changes denoting where the text has been changed; the other a clean version (uploaded as the manuscript file).

Additionally, to enhance the reproducibility of your results, PLOS recommends that you deposit your laboratory protocols in protocols.io, where a protocol can be assigned its own identifier (DOI) such that it can be cited independently in the future. For instructions see http://journals.plos.org/plospathogens/s/submission-guidelines#loc-materials-and-methods

We hope to receive your revised manuscript within 60 days. If you anticipate any delay in its return, we ask that you let us know the expected resubmission date by replying to this email. Revised manuscripts received beyond 60 days may require evaluation and peer review similar to that applied to newly submitted manuscripts.

[LINK]

Sincerely,

Ronald Swanstrom

Associate Editor

PLOS Pathogens

Susan Ross

Section Editor

PLOS Pathogens

Kasturi Haldar

Editor-in-Chief

PLOS Pathogens

orcid.org/0000-0001-5065-158X

Grant McFadden

Editor-in-Chief

PLOS Pathogens

orcid.org/0000-0002-2556-3526

Reviewer's Responses to Questions

**Part I - Summary**

Reviewer #1: This reviewer was looking forward to evaluating this manuscript as corresponding author usually does very nice work. However, this reviewer feels that current manuscript has many claims that are not fully supported by data presented. The major claim (for this reviewer) being the linkages amongst: (i) glycolytic activity; (ii) membrane tension; and (iii) HIV fusion.

It is my opinion that authors have used a sub-optimal system to link the HIV biology with glycolytic activity or membrane tension. Incorporation with relevant T cells data on key aspects would have provided huge amount of reassurance.

In terms of linkage between glyolytic activity and membrane tension, the interesting part begins when FliptR dye is used, BUT the data appear to fell short on definitively making a strong link between glycolytic activity [metabolism] and membrane tension (at least for this reviewer).

There is large amount of errors within the manuscript (in particular with figure and supplement figure referrals), which unfortunately make this assessor has (perhaps unnecessary) doubt on some of the work.

Overall, authors have used a lot of cool and exciting techniques in this paper, BUT the authors seem to not have careful considered the limitations of various technique used (at least this is the interpretation of this reviewer). Claims being made in the paper are NOT readily justified in my view. There are fair bit contradictions presented in this revised manuscript that I evaluated (see below).

Overall, authors try to look at the impairment of metabolic effect might have on virus entry. Authors claims that there is a link between metabolic state of the target cell and the ability for the virus to infect. If one takes a simple step back, such premise is not surprising at all. Virus (by definition) hijack host cell system to achieve its own gain. From the point of virus entry, establishment of infection, genetic materials and protein productions, to the stage of assembly of virus, ALL rely on the host cell to be somewhat healthy to begin with - ie metabolic active to generate energy to the rest of the cells, thereby invading virus can take advantage of.

On the other hand, if authors are trying to show a particular subset of metabolic events is purposely hijacked or manipulated by viruses, it would be extremely interesting. To do that, one should demonstrate any observed effect is related to such subset of metabolic events, but the manipulation was NOT affecting the global process (issue of pleiotropic effects).

Unfortunately, in this manuscript, authors have chosen to use a very blunt instruments (in terms of biological systems [TZM-bl] as imaging tools are top notch) to demonstrate this point. Authors appear to over-push the imaging technology fronts that this lab is good at, without carefully consider all the limitations (at least this is this assessor's opinion). Please see specific comments below.

Having said that, all of these imaging are state-of-the-art techniques. Some time it is less clear for this reviewer whether authors are show-casing these techniques or answering a specific question for this retrovirus.

Key points in summary include the following:

2DG impairing glycolysis, then affect lipid membrane function in association of cholesterol. This could simply be a reflection of a dynamic process in lipid membrane (that requires energy) being important for virus entry?

My own interpretation is that affecting glycolysis could simply be that target cell for infection require the membrane to be healthy (to have dynamic lipid movement) to enable virus to enter. in that sense, if the majority of metabolic process are in idle, it might not be surprising for authors to see these outcomes.

cholesterol-lowering medication in HIV infection is a pretty big stretch by the authors as far as this reviewer is concerned.

From author summary, ‘regulating cholesterol’ was mentioned. It might not be related to cholesterol regulation on plasma membrane. It is more likely related to maintain dynamic movement of plasma membrane. Observed cholesterol supplement effect could be secondary effect of cholesterol uptake that permits / restores a more dynamic movement of plasma membrane under these 'stress' conditions.

This assessor has different interpretation from authors on their comments on metabolism, immunity, and viral infection. Authors cited Buck 2017 and Pallett 2019, and I have gone in to read those papers. My personal read is that metabolism needs to be active to allow cell to perform. Likewise, if a cell is dormant and is not doing a lot, it will not be any good to serve immune function to counteract pathogens. Similarly, a less active cell will not be effective for pathogens to infect that lead to production of progeny pathogens. This is different from a sick cell that can still produce energy source, but innate immunity is down. A key issue perhaps is how much the host cells are in idle (80% functional vs 20% functional in energy production). This relationship is particular important for viruses, as viruses rely on host cell machineries to achieve its own gain. Unless the target cells and host cells are 'fairly' active in generating resource for pathogens to take advantage of, there is not a lot of use for viral pathogens. I think we all agree that biological process requires energy in one form or the other.

The entire infection process requires energy, and viruses DO NOT bring in its own energy production source. The simple explanation (that authors observed) is an effect at the entry stage of virus-cell interaction, which reflects the very first step of energy expenditure process that virus relies on the host cells - ie entry.

Authors described hemi-fusion event and membrane curvature in an attempt to link cholesterol in HIV infection. However, the system that the authors used also including the VSV-G HIV system, which is unrelated to HIV Env. VSV-G mediated endocytosis entry is largely unaffected by cholesterol (at least with particle-associated cholesterol, and I recalled at least 3 labs have independently reported this). On that note, making it more difficult for the authors to link their story with cholesterol specifically. Again, to me, the phenotype is more likely a general impact on metabolism. More concerns about hemifusion events are described below.

With PLoS Pathogens, there is NO limit on numbers of figures or word count. There are many supplement figures in this paper should be incorporated as main figures. Authors should correct this before publication. Authors claim current format is intended to be helpful for readers, and I do not share their enthusiasms. It looks though other reviewer is feeling the same. Current formats appear to be something originally submitted to another journal (eLife for example), and this format is not helpful in my opinion. in fact, there are A LOT of errors within this current manuscript that is very damaging to the paper (see specific below).

Reviewer #2: Coomer et al. report a beautiful study of the importance of the properties of the target membrane on HIV fusion, and its link to the metabolic state of the cell. They find that reduction in glycolysis removes cholesterol from the plasma membrane, which increases membrane tension. This increase in tension is less favorable for HIV fusion. As a result, virions entering at the plasma membrane stall at the hemifusion stage, as evidenced by observation of lipid mixing in the absence of content release. While the importance of target cell metabolic state and the physical properties of the membrane have previously been reported as important to HIV fusion, this study makes clear the inter-relatedness of these previous observations and generates a more unified understanding.

As one has come to expect from the Padilla-Parra laboratory, the authors use quantitative optical microscopy approaches, and a variety of fluorescence-based assays, which appear to be rigorous and well executed. By my evaluation, the previous reviewers were appropriately thorough; and the authors have nicely addressed all major points raised. I therefore have only minor comments.

Reviewer #3: Overall, I thought this was an interesting paper. I think the assays that the authors have used are clever and allow them to begin to answer very difficult questions in our field. I particularly liked the coupling of the FLIM images with infectivity assays and fusion reporters in non-treated cells, since as the authors mentioned, the experiments are internally controlled and the metabolic pathways of the cells are not perturbed. At times, I wish the data they presented were a bit cleaner, but the statistics look good. The biggest frustration I had with this paper was the number of errors in labeling and writing. These errors made important parts of the paper very difficult to decipher, and at times, I was left wondering if I interpreted their errors correctly and was understanding the data properly. To be honest, I am not someone who often picks up on these types of errors, so the fact that I found so many suggests that it’s a significant problem. The authors need to read this paper VERY carefully and correct all errors.

**Part II – Major Issues: Key Experiments Required for Acceptance**

Reviewer #1: Figure 1 TZM-bl. TZMB1 spreads out very well, and they are excellent for cell morphology in terms of imaging. TZM-bl are also very tough-resilient, and they are not real cells for HIV infection. MT4 is good, but still limited. For the claim to be linked with HIV, it would be necessary for author to repeat figure 1A with PBMCs (2-DG dose response) along with cell viability. Maximum impact is 50% reduction of infectivity with VSV-G pseudo typed HIV. I wonder what the impact it would be using PBMCs and R5-tropic (or even transmission strain HIV) to infect. The key concern is that the 2-DG treatment described was way too tough to the host cell to begin with. By impairing the major metabolic processes to the host cells, these cells are essentially too weak (perhaps even dying) to do anything, therefore not able to support HIV infection as consequence. If authors insist on their claim with HIV, I think it is only fair that they have appropriate T cell infection data.

On the other hand, if authors were changing their claims, experiments in T cells might not be needed. More specifically, the beautiful imaging data are excellent. To me, they demonstrate to me that energy requirement is needed to have a dynamic membrane (including cholesterol movement). Such dynamic movement of plasma membrane as well as membrane related endocytosis process are fundamental for particles uptakes. What authors are seeing in both HIV particles in TZMBL and VSV-G HIV in TZMBL are reflection of that. This could well be a general principle for particles uptake (including many viruses and perhaps even exosomes). Such general principle could be best demonstrated using various viral envelope pseudotyped particles (Zika Env, SARS Env, Ebola Env and VSV-G to just named a few) plus a number of other relatively easy to image fibroblast typed cells. Authors can even use the exact same imaging system that they have right now.

Legends in Figure supplement Figure 1A are not adequate. Which one is HIV-1(VSV-G) or HIV-1(NL4-3)? Is VSV-G on top of NL4-3 or VSV-G on top of Envelope negative HIV? In fact, the presentation of supplement Figure 1A is not up to usual standard of this team. I would imagine authors should synchronise all fonts and font size throughout all figures (including supplement) to deliver their points to reviewers in order to help to demonstrate their case effectively, as well as best to illustrate the quality of works being done. Presentation of supplement figures are no less important from the main figures.

With Figure 1 supplement figure 1B, is that laconic or perceval? Similar question with supplement figure 1d. what is on the label with this supplement figures vs what is written in the supplement figure legends are completely opposite - which one is the truth? Same question for Supp Fig1E vs Supp Fig1F. Noticed author rebuttal, but I am not sure error has been corrected - at least with the version I have.

Authors should have better explain what the oligomycin control is used for, and what that meant. Currently, it is very inadequate in this assessor's view

With the schematic on Fig 1C and the ACTUAL data in Fig1 Suppl Fig1F-H, authors seem to suggest the metabolic state of the cell are significantly affected by 2-DG. With this in mind, would it not true a simple explanation being that the 2-DG treated cells are now so weak to produce energy source that virus can no longer hijack the cellular machineries to establish infection? A trivial and simple explanation. An important character of virus being that it relies on host cell to replicate, and the key issue for that being virus cannot generate its own energy source and metabolically inert.

Arguments made by authors from lines 142 to 152 are not interpretable given the conflict of figure 1 supplement figure 1B-E legends

With limitation / conflict of figure 1 supplement figure 1B-E legends, paragraph from line 153 to 161 are not easy to understand

At the first glance, data in Fig 2 are really cool set up and experimental approaches. However, the choice of using TZM-bl cells appears to be a waste opportunity for this assessor. Only if authors apply these in more relevant cells. Just because TZM-bl are easy to image, it does not mean that TZM-bl is best for the current biological question in hand (see below with more details or above for tailoring claims against data presented).

With Vpr-b-lam assay, the enzyme b-lam is from virion, substrate CCF2 is passively introduced (does not require metabolic events of the target cells). There are a lot of HIV ‘fusion’ events seen in 2B. What is the background level of b-lam activity with this batch of TZM-bl cells of the authors. TZM-bl or HeLa are well known engineered cells, TZM-bl (and HeLa) can vary from one lab to the next, early day cell engineering often utilise plasmid contain ampicillin resistant gene (blaTEM-1). It would be great (in fact important) for authors to confirm their TZM-bl cells to have low background of b-lam gene products (without HIV infection) to reassure their observations of red (presumed fusion events) are indeed related to HIV entry (therefore releasing b-lam from Vpr-conjugation). This assessor has seen data of different TZMbl used for b-lam entry assay with significant different levels of backgrounds. Authors should clarify this. There is a scale bar in second panel of 2b, but actual scale was not described in the legend.

With data in Figure 2, authors suggest that low metabolic activities are related to lower fusion events for HIV entry. Here is an alternative interpretation. Assuming that there is background b-lam issue with the TZMBl cells being used, therefore it is possible that lower metabolic activity would lead to less endogenous b-lam being expressed, hence, loading of CCF2 dye will give rise to less portion of red (presumed fusion positive cells), while adding HIV into these cells will lead to smaller fraction in terms of changes to ‘presumed fusion positive states’. Authors could still be correct, but these alternative possibilities must be eliminated. Also, TZM-bl is great for endocytosis mediated entry (which are also supported by data presented by this team). At the end of the day, best possible controls please;

From the paragraph starting in line 177, authors indicated that treatment of 2-DG does not affect viability of cells based on propidium iodine staining. Two questions emerge: (i) is propidium iodine staining the best assay for this? As described above, what if the cells have become SO fragile / weak that are no longer able to support most of their biological activities due to not making enough ATP (but they are not yet dead). As a consequence, much of the mechanism viruses rely on to infect the host are no longer available. These top down overall effects on shutting down metabolic process would complicate interpretation. In the case the propidium iodine staining, the cells HAVE to be dead in order for the dye to go in, hence it is not fair to say that 2-DG does not have toxic effects to the cells. These cells are likely to be so weak but not dead JUST YET – would you not consider this cytotoxic effect or pleiotropic effect? (I would); (ii) Also, HeLa based cells (including TZMbl) are generally MUCH tougher than many other cell types in the lab, making the thresholds of picking up propidium iodine stains much higher. This reviewer would suggest that parallel experiments should be done on primary T cells to begin with, in particular given authors’ claim on potential clinical application by extrapolating these observations on cholesterol lowing medication in HIV infection (assuming HIV linkage remains a key claim from the reviewers).

Another way to look at it would be dynamic movement of lipids (including cholesterol movement in PM) requires energy. What authors have observed (impacts on HIV entry in TZMBl cells) is simply reflection of this defects. In the other word, the phenomenon with HIV entry is a real out of plasma membrane dynamics, but not a SPECIFIC event that is unique to HIV. Perhaps consider showing this as general principle for various particles uptake (including exosomes across larger number of fibroblast cell types? General principle seems to be more important to me than HIV infection.

By the way, if authors feel that they are correct that cholesterol lowering drug can impact on HIV replication in a way that they suggest, perhaps they should simply do that experiment using primary cells and physiological relevant dosage of cholesterol lowering drugs to see whether HIV entry is affected based on their hypothesis. If their claim is true, it will be a much stronger paper using these high-end imaging approaches to show the HIV linkage and increase accuracy of their prediction / hypothesis.

For information in line 182, it has been reported that with the endocytosis mediated HIV entry, the cholesterol (at least particle-associated cholesterol) in VSV-G pseudotyped HIV is NOT important for virus entry (reported by multiple labs). On the surface, current interpretation of the authors is not in line with what the field knows. However, this reviewer agrees that whether overall cholesterol levels in the host cells is important for VSV-G pseudotyped HIV can affected endocytosis mediated entry is not known. The main reason (in my opinion) such information is hard to emerge as the pleiotropic effects of such manipulation on the entire cell, making it impossible to properly interpret some of those experiments. As far as I know, there is NO simple / clean way to manipulate cholesterol on PM without affecting other parts of the cells. This assessor feels that the authors are facing similar problems with their claims here on metabolic activity. Unless there is a way to ISOLATE a specific effect on cholesterol regulation at the plasma membrane alone from manipulate metabolic process, proper interpretation would be difficult.

Would have been good if there is a figure number (1, 2, 3, …) in all of the figures and supplement figures. While authors might have label individual figure file with such info, this information is missing in the merged file. This is particular important that all supplement figures are all group together, but there is NO supplement figure 2. For example, there are Figure 1 supplement 1, then jumped to figure 3 supplement 1 and figure 3 supplement 2. This information was not obvious until this assessor has gone through the legends of supplements – unnecessary confusing and annoying! Anything helps reviewers to better understand the paper is a good thing (in my opinion). On the same points, authors are extremely careless in describing and citing their figure number in the text. Very often, author mis-referred to figures, and I was not able to find all figures mentioned. Given this is already a revision, it is a bit concerning.

Line 186 on (supplement Figure 3B) is not clear. Do author mean Figure 3 supplement 1B or Figure 3 supplement 2B.

If authors refer to Figure 3 supplement 1B for comments in Figure 3 supplement 1B, yes, I can see that is statistically significant with *. However, judging from the error bar, it is almost overlapped, and the p-value must be just under 0.05. What if you do that experiment with more biological relevant cells for HIV – ie PBMCs, where entry numbers will be low, and more sensitive to damage by these types of treatments. I wonder what the p value be then.

Label in Figure 3 supplement 1C are WAY too small. Again, line 191, wording of ‘supplemental Figure 3C’ was not clear enough for Figure 3 supplement 1C or Figure 3 supplement 2C. Reviewers and audience should not have to guess or to assume.

With figure 3 supplement 2c, co-labelling experiment to look at hemifusion using iGFP and DiD dye. Authors should be congratulated on doing these experiments, as they are tough experiments. However, I would suggest the N values of these are limited. Notwithstanding that these are very tough experiments, but from the vantage point of reviewer, I would ask the question whether these numbers are sufficiently significant to demonstrate the claim, rather than the level of difficulty of these experiments themselves. Also, authors use TZM-bl system. Why not primary T cells or even T cell line? I seem to recall live imaging of T cells have been done by highly respectable scientists at Oxford some years ago. Would they not be near by the authors’ lab? At least you are all at same University. Furthermore, the upward time (shown) for imaging is 10min that have been done by this team. Understand the nucleus are huge in T cells, BUT since the question is about PM based entry anyway, there will not be a major issue. T-cell experiments would be much more convincing for claim linking with HIV infection.

No question that there are many very cool and advance technologies throughout utilised by the authors, but cart should not come in front of the horse. Even with large number of COOL technologies, if biological question cannot be answer properly in the stated claim with the data presented, arguments will fall down. The only exception would be if that is a technological paper to begin with, but this is not the case here. I feel the authors 'overplay' their technology card

From lines 202 to 228, there are A LOT of works here. These are also difficult experiments. I also understand that spin-inoculation will help the number to go up for measurement purpose. Personally, I would have been much more interested to see the ‘Not-Spun’ virus infection experimental data, bright green of Figure 3 supplement 2C. From the text, authors seem to indicate that the data from Main Figure 3C are for non-SPUN virus SVT, perhaps it will be good to remove SPUN virus data from Figure 3 supplement 2 to avoid confusion.

Line 205 ‘Supplemental figure 3D’ meant ‘Figure 3 supplement 1D’ as seen with legend?

Line 215 ‘Supplemental figure 3E’ meant ‘Figure 3 supplement 1E’ as seen with legend?

In ‘Figure 3 supplement 2C’, the distinct categorisation between ‘hemifusion’ and ‘colour separation’ are not clear. According to description on line 213-217, there are two types of DiD dye disappearance events. One is ‘near-asymptotic loss’, while the other one is slow DiD decay. Authors clearly indicated that 7% of rapid decay (colour separation) was previously reported in Vega et al 2011. If I am not mistaken what I have read in the Vega et al paper, the fast decade of DiD (7%) in this case, was referred to be evidence of ‘hemifusion’ event in the last paper (see Vega et al 2011, under Section of ‘Results’, Sub-Section of ‘Blocking HIV-1 uptake by a dynamin inhibitor …’, the last two sentences of Paragraph, and I quote

‘These findings are consistent with the notion that the slow loss of DiD fluorescence is caused by dynasore, whereas the fast DiD decay reflects transfer of viral lipids into the plasma membrane mediated by Env. By definition, lipid mixing in the absence of content release corresponds to a hemifusion phenotype.

If these rapid DiD lost event WAS related to hemifusion event before in the Vega paper, why would authors CHANGE the definition NOW that in this MANUSCRIPT that ‘Hemifusion is related to the slow release of DiD dye?

Authors MUST clarify the distinction between hemifusion vs colour separation.

Also, given data in Figure 3 supplement 2C that authors clearly have the capacity to distinct different categories of dye disappearance (endosomal fusion, hemifusion, colour separation and plasma membrane fusion), author should (and perhaps MUST) present their main Figure 3C data (with vs without 2DG treatment) that way (ie with these 4 categories - endosomal fusion, hemifusion, colour separation and plasma membrane fusion). These would be MUCH MORE meaningful. It is VERY puzzling that authors have not done that AND make the assumption and I quoted

‘it is extremely likely that the observed hemifusion arrest by 2-DG explicitly occurs in the plasma membrane’ (see line 227-8). This assumption does not seat well with this reviewer. Would authors like to see other manuscript present their major claim of the paper (as seen in abstract and title) basing on an assumption? This is particularly puzzling given the data are within the reach of the authors, and the experiments are high quality.

On line 223, authors suggested there is (supplement figure 4, related to figure 3). Again, the labelling of figures by authors are poor. I could be wrong, but I was not able to find a supplement figure that is related to spun and non-spun data.

On line 224, authors claim that PM fusion occurs in a high probability. However, the data in Figure 3 supplement 2 C , there are more endosomal fusion events.

Under section of ‘Single-Cell Metabolic State is linked to … …’, line 230, authors use filipn stain to quantify cholesterol. Stain-ability by filipn is a very indirect way to quantify cholesterol on PM. Why not use mass spec by isolating the membrane fractions?

Lipid membrane in PM is dynamics, binding of cholesterol with filipin will OF COURSE affecting PM architecture. This reviewer disagrees with comments made in line 237 by authors. I am surprised authors will cite a book chapter published in 2019 (which I do not have access due to non-open access to verify the authors’ claim from this book). I however do notice that authors have ignored large number of papers and reviews published on lipid rafts and filipin in the past 20+ years. A simple search would have revealed filipin effects on plasma membrane of cells are cell type dependent and concentration dependent. By searching the entire submitted manuscript, authors did not indicate the flipin concentrations used, nor provide control data showing their flipin labelling does not adversely affect the host cells in their hands.

The MBCD treatments described between lines 239 and 246 are irrelevant to me. In the presence of mixing filipin as a stain, then MBCD, there will be too many effects on the PM, making things way too complicate to interpret. In fact, MBCD will cause cholesterol shuttling between PM and extracellular environment much more rapid.

In my opinion, exogenous cholesterol supplement experiment is poorly designed. When exogenous cholesterol is introduced into cell, it will promote cholesterol uptake by cells. If the overall metabolic events of the cell are affected, and impacting on normal cholesterol synthesis then PM targeting, exogenous cholesterol can easily be picked up by host cells from extracellular environment. As a consequence, virus particles such as HIV can readily be picked up – Brugger et al 2006 PNAs already described HIV essentially is a specific type of rafts. Consequently, data in Figure 2C are therefore completely expected for this reviewer – but not for the reason authors thought or claimed.

Authors indicated there is figure 4D (line 253), but this assessor cannot find figure 4D

This assessor does not know enough about the FliptR dye. However, would one approach be enough to establish an important claim between metabolic events and membrane order? Notwithstanding some of the limitation of the Laurdan, why not use it to provide additional data to strength the claim? In particular, potential coupling with PALM/STORM on CD4 as others have done on LAT (Owen 2012 Nat Comm).

Reviewer #2: (No Response)

Reviewer #3: 1. I believe that Supplemental Figure 1 is still not correct. I think Figures 1C and 1E need to be flipped — they do not correspond to the graphs in 1B and 1C. Also, the figure legends for 1B and 1C do not correspond to the labels in the figure. Especially since the two metabolic reporters show opposite trends, these errors made interpreting Main Figure 1D very hard. However, I think I figured it out and that the data in the main figure make sense.

2. I think the author’s should give better descriptions of their reporter assays. Most HIV readers will understand the assays, but the article should be accessible to others. For example, in Figure 2A, they show the chemical reaction that is catalyzed but don’t explain why a vpr-blam fusion would be a marker for viral entry. The descriptions should be brief but the reader shouldn’t have to go to other references to understand the basis of the assays.

3. With regards to the sometimes poor overlap of cells in the different images in Figure 1 and 2 that the previous reviewer mentioned. The authors arguments make sense, but would it be possible to include transmission images of the cells before and after? It might make it more obvious that the same cells are being imaged and how they have changed. Also, in Figure 2C, it would be helpful if there were images of regions with lower cell density so that the cell identities would be more obvious.

4. Why does the image in Figure 4C w/ 400ug/mL of cholesterol have so many non-infected cells? It appears from the graph that the infection rate should be much higher.

5. The authors should consider moving some of the supplemental data into the main figures. Specifically, I found the Flipper-TR data to be clean and convincing and think it should have a more prominent place in the paper.

6. I think that the authors need to address the effects of cholesterol on the endosomal membrane. Even in their “no spin” experiments, a significant portion of fusion events appear to go through the endosome. If they’re trying to correlate infectivity with membrane effects then they need to show convincing data that perturbations that they make to the plasma membrane have similar effects on the endosomal membrane.

**Part III – Minor Issues: Editorial and Data Presentation Modifications**

Reviewer #1: No additional comments

Reviewer #2: It is worth commenting on whether glycolysis is likely to affect plasma membrane composition beyond cholesterol.

Why is Figure 1C largely repeated in Figure 2A? This seems unnecessary.

At several points throughout the figures (e.g. Figure 2B) histograms are presented along side example images. The axes of the histograms are not labeled, and they are not defined in the legend, making them difficult to interpret for the unfamiliar reader.

On line 253, the authors refer to Figure 4D. Figure 4 only has panels A through C.

In the model in Figure 6 the hemifusion arrested state is depicted in an unusual way, with the inner leaflet of the viral membrane disrupted. Typically, the inner leaflets are thought to be intact in the hemifusion state. Was this intentional? If not, I would suggest drawing this in the conventional way so as to avoid confusion.

Reviewer #3: 1. The G and H figure legends are also swapped in Supplemental Figure 1. In the legend it says that the G graph corresponds to 2-DG, while in the figure it is labeled Oligomycin.

2. Supplemental Figure 2 is mislabeled Figure 3 in the figure legend.

3. I’m assuming in Supplemental Figure 3C that this is not “%” but “fraction”? 0.25% seems very low.

5. I believe the MT-4 panel of cell images is missing in Figure 4.

6. Some of the labels are cutoff in Supplemental Figure 4A.

7. The authors should label the graphs in Supplemental Figure 5F.

8. Supplemental Figure 6 is mislabeled Figure 5 in the figure legend.

9. The authors should consider labeling more explicitly the figures themselves. It’s frustrating to have to constantly dig through figure legends to understand what the graphs are showing.

PLOS authors have the option to publish the peer review history of their article (what does this mean?). If published, this will include your full peer review and any attached files.

Reviewer #1: No

Reviewer #2: No

Reviewer #3: No

---

## [Decision Letter · Decision Letter 1]

27 Jan 2020

Dear Dr Padilla-Parra,

We are pleased to inform you that your manuscript 'Single-cell glycolytic activity regulates membrane tension and HIV-1 fusion' has been provisionally accepted for publication in PLOS Pathogens.

Before your manuscript can be formally accepted you will need to complete some formatting changes, which you will receive in a follow up email. A member of our team will be in touch within two working days with a set of requests.

Best regards,

Ronald Swanstrom

Associate Editor

PLOS Pathogens

Susan Ross

Section Editor

PLOS Pathogens

Kasturi Haldar

Editor-in-Chief

PLOS Pathogens

orcid.org/0000-0001-5065-158X

Michael Malim

Editor-in-Chief

PLOS Pathogens

orcid.org/0000-0002-7699-2064

Reviewer Comments (if any, and for reference):

Reviewer's Responses to Questions

**Part I - Summary**

Reviewer #1: A very comprehensive revision, and it is certainly one of the more detailed revisions I have seen in recent time. Thank you.

Reviewer #3: This paper describes a study that explores an important problem in our field. The experiments in the paper are well-done, and the conclusions are novel and very interesting. In addition, the authors have addressed my concerns in this revised manuscript.

**Part II – Major Issues: Key Experiments Required for Acceptance**

Reviewer #1: Not Applicable

Reviewer #3: None

**Part III – Minor Issues: Editorial and Data Presentation Modifications**

Reviewer #1: Not Applicable

Reviewer #3: None

PLOS authors have the option to publish the peer review history of their article (what does this mean?). If published, this will include your full peer review and any attached files.

Reviewer #1: No

Reviewer #3: No

---

## [Editor Report · Acceptance letter]

14 Feb 2020

Dear Dr Padilla-Parra,

We are delighted to inform you that your manuscript, "Single-cell glycolytic activity regulates membrane tension and HIV-1 fusion," has been formally accepted for publication in PLOS Pathogens.

Best regards,

Kasturi Haldar

Editor-in-Chief

PLOS Pathogens

orcid.org/0000-0001-5065-158X

Michael Malim

Editor-in-Chief

PLOS Pathogens

orcid.org/0000-0002-7699-2064